# A dual fluorescent-Raman bioorthogonal probe for specific biosynthetic labeling of intracellular gangliosides
Mana Mohan Mukherjee [1], Matthew D. Watson [2], Devin Biesbrock[1], Lara K. Abramowitz[1],
Steven K. Drake [3], Jennifer C. Lee [2] & John A. Hanover [1] ✉

Gangliosides are sialic acid-containing glycosphingolipids integral to the cell membrane, and they are particularly abundant in the nervous system. Aberrant ganglioside metabolism contributes to pathological conditions, including neurodegenerative diseases, lysosomal storage disorders, and cancer. A critical precursor for sialic acid biosynthesis is *N*-acetyl-D-mannosamine (ManNAc), which can be epimerized from the corresponding UDP-GlcNAc or exogenously supplied through ManNAc derivatives. Currently, tools to visualize and detect gangliosides are very limited and non-specific. Here, we describe a dual fluorescent and Raman-active ManNAlk derivative, phenanthrene-9-Pr$_4$ManNAlk (MM-JH-2), capable of one-step selective labeling of gangliosides in cells. This modified ManNAlk derivative produces a biologically unique Raman spectral signature, which arises from the carbon-carbon triple bond augmented by conjugation to a fluorescent phenanthrene moiety. Raman maps generated using the alkyne stretching frequency indicate a distribution of MM-JH-2 overlapping with intracellular membrane lipids. Using confocal fluorescence imaging, the cellular transport of labeled gangliosides was tracked. Notably, MM-JH-2 can differentiate between cells that differ in ganglioside biosynthetic flux, such as malignant and nonmalignant cells, as well as distinguish between B cells and T cells. Thus, MM-JH-2 is a next-generation metabolic chemical reporter (MCR) that is Raman-active, fluorescent, and can be broadly applied to cellular studies investigating ganglioside biosynthetic flux.

Gangliosides are a class of sialic acid-containing glycosphingolipids that are particularly abundant in the brain and the nervous system. Gangliosides participate in the maintenance and mending of neuronal cells, memory formation, synaptic transmission, regeneration of neurons, axon stabilization, modulation of signal transduction pathways, and both intracellular and intranuclear calcium homeostasis[1]. During brain development, ganglioside expression undergoes qualitative and quantitative changes[2]. Alterations in ganglioside metabolism contribute to rare hereditary paraplegia, intellectual disability, Alzheimer's disease (AD), Parkinson's disease (PD), insulin sensitivity and diabetes, bacterial toxin susceptibility, and severe lysosomal storage diseases[2,3]. Further, ganglioside-targeted autoimmunity is responsible for certain forms of Guillain–Barré syndrome, a common cause of acute flaccid paralysis[2,3]. In addition, the function of gangliosides in regulating growth factors and receptor tyrosine kinases contributes to their involvement in carcinogenesis. Ganglioside expression is altered in many cancers, and is a target for cancer immunotherapy[4–6]. In 2015, the monoclonal antibody against GD2 ganglioside, dinutuximab, was approved for treatment of high-risk neuroblastoma[4,5], and there are ongoing global phase-II randomized trials of anti-Globo H as a vaccine against metastatic breast cancer[4,5].

Since the first isolation of gangliosides from the human brain by Ernst Klenk in 1942[7], research has focused on structural analysis, biosynthetic pathways, and subcellular localization[8]. However, analysis of gangliosides remains difficult due to their amphiphilic nature and inability to withstand traditional purification techniques and washings. Techniques for capturing ganglioside complexes in their native environments would represent a powerful tool for characterizing these labile complexes. Currently, antibodies against various gangliosides, designated as stage-specific embryonic antigens (SSEAs), have been widely used to characterize the ganglioside profiles of embryonic stem cells (ESCs). However, these antibodies often

[1]Laboratory of Cell and Molecular Biology, NIDDK, NIH, Bethesda, MD, USA. [2]Laboratory of Protein Conformation and Dynamics, NHLBI, NIH, Bethesda, MD, USA. [3]Critical Care Medicine Department, Clinical Center, NIH, Bethesda, MD, USA. ✉e-mail: johnh@bdg8.niddk.nih.gov

display cross-reactivities[9]. Thus, ganglioside profiles in human ESCs and differentiated derivatives are more accurately delineated by systematic surveys using matrix-assisted laser desorption/ionization time-of-flight mass spectrometry (MALDI-TOF-MS) and tandem mass spectrometric analysis. However, neither this technique nor thin-layer chromatography (TLC) analysis can provide spatial information about cellular distribution or real-time molecular mechanisms of ganglioside interactions. Rather, these techniques rely on analysis of bulk or fractionated cell components following extraction.

Here we demonstrate a bioorthogonal chemistry[10–12] derived ManNAlk analog that is capable of the one-step metabolic labeling of sialic acid to catalyze the biosynthesis of gangliosides. Bioorthogonal chemistry traditionally involves two steps. First, metabolic incorporation of the unnatural monosaccharide, and second, an azide-alkyne cycloaddition "click" reaction (Cu-catalyzed or Cu-free) with a fluorophore or affinity tag[12,13]. Despite their high efficiency in vitro, it can be challenging to drive click reactions to completion in cellular environments due to slow ligation or side reactions with various functionalities within the cell. Therefore, a one-step labeling strategy relying only on enzymatic transfer, and circumventing the click reaction, increases sensitivity[14,15]. Further, direct fluorescent labeling of glycans is particularly useful for high-resolution live-cell imaging of larger regions of interest in complex biological systems.

The biosynthetic precursor of sialic acid is ManNAc, which can either be exogenously supplied or biosynthesized endogenously from N-acetylglucosamine (GlcNAc) or UDP-GlcNAc within the cytoplasm[16]. In a group of mammalian cell lines, per-O-acyl protected ManNAc analogs were incorporated with almost a 1000-fold increase in efficiency compared to free monosaccharide counterparts. The lipophilicity of these analogs is increased via protection of the sugar skeleton hydroxyl groups with lipophilic esters (Ac$_4$ManNAc, Pr$_4$ManNAc, Bu$_4$ManNAc)[16]. Once inside the cell, these ester functionalities are removed by broad-spectrum esterases to generate naked/core ManNAc that enters the targeted biosynthetic pathway. However, longer-chain ester protecting groups increase the cytotoxicity of the compound. Bu$_4$ManNAc, for example, causes significant cell death at high concentrations (500 µM)[16]. Fortunately, when using per-O-acyl protected ManNAc analogs, concentrations of less than 100 µM are generally sufficient for ample metabolic incorporation, nutrient flux, and glycan labeling.

The alkyne functionality in ManNAlk carries a distinctive vibrational spectroscopic signature. The C≡C stretching mode produces a Raman band around 2200 cm$^{-1}$. This is located within the "cellularly-quiet" region of the Raman spectrum, from approximately 1800 to 2800 cm$^{-1}$, in which there is no spectral interference from endogenous biomacromolecules[17,18]. Thus, the carbon-carbon triple bond affords background-free detection and unambiguous assignment of the sugar moiety[17]. Another advantage of Raman spectral imaging is that it allows multiplexing so that the localization of other biomacromolecules (e.g., nucleotides, proteins, and lipids) can be mapped simultaneously without the addition of exogenous probes. Previous studies have shown the use of alkyne-functionalized unnatural amino acids as Raman probes in cells[19,20]. Despite the absence of interfering Raman signatures, spontaneous Raman imaging using terminal alkynes can often be challenging, depending on the relative intensity of the alkyne stretching band[17]. However, sensitivity can be improved by chemical modifications such as simple aromatic substitution on the terminal alkyne, which increases signal intensity more than 10-fold[17].

We sought to leverage the advantages of both Raman and fluorescence imaging by synthesizing a dual fluorescent-Raman ManNAlk derivative for specific labeling of gangliosides in mammalian cells. Here, we report MM-JH-2 as a next-generation MCR where the large hydrophobic substitution on the glycan assembly directs the compound towards glycosphingolipids. The fluorescent property of the conjugated phenanthrene[21], along with the Raman activity of the non-terminal alkyne, can be used independently for cellular distribution and metabolic studies. This modified ManNAlk derivative, MM-JH-2, is fluorescent upon blue-light excitation (405 nm), and its Raman spectrum can be measured using longer wavelengths of light (eg. 514 nm).

## Results

### Synthesis of MM-JH-2 and fluorescent labeling of cells

Taking advantage of the utility of neutral ManNAc derivatives in metabolic oligosaccharide engineering (MOE), for the monitoring of sialic acid metabolism, we designed and synthesized a fluorescent ManNAc derivative, MM-JH-2, for one-step direct labeling of sialo glycoconjugates. MM-JH-2 incorporates a bioorthogonal Raman active alkyne moiety (C≡C), as well as a fluorescent conjugated phenanthrene. Briefly, D-mannosamine hydrochloride (1) was converted to ManNAlk (2)[22] followed by Sonogashira coupling with 9-iodophenanthrene to afford phenanthrene-9-ManNAlk (3) in high yield. The hydroxyl groups were masked with propionic ester to produce phenanthrene-9-Pr$_4$ManNAlk (MM-JH-2, 4, Fig. 1A). Formation of the target compound, was confirmed by high-resolution mass spectrometry (HRMS) ([M+Na]$^+$ calculated for C$_{37}$H$_{41}$NO$_{10}$Na 682.2628; found 682.2629) and detailed NMR analysis of both the anomers which contained five carbonyl carbon peaks in the $^{13}$C NMR spectra (four -OC=O and one for -NHC=O), and peaks for the aromatic functionality in both the $^1$H (8.7–7.5 ppm) and the $^{13}$C (140-110 ppm) NMR spectra (Supplementary Figs. 1 and 2). Fluorescence excitation and emission spectra of MM-JH-2 reveal a broad excitation band between 350 and 425 nm with an emission maximum at 437 nm, which is consistent with the phenanthrene functional group (Fig. 1B).

With this pure compound in hand, the labeling efficiency of MM-JH-2 was investigated in a variety of mammalian cell lines, including HeLa, SH-SY5Y, LA-N-2, HEK293T, NIH 3T3, AML 12 and LecCHO. Cells were treated with differing concentrations of MM-JH-2 for 72 hours (Fig. 1C and Supplementary Figs. 3 to 8), similar to previous studies assessing ManNAc derivatives[23]. The compound was imaged in cells using 405 nm laser excitation with emission intensity normalized to SYTO Deep Red Nucleic Acid Stain. The mean fluorescence intensity increased with increasing concentration of MM-JH-2 for all cell lines. Detection limits varied somewhat between cell lines, ranging from 20–25 µM treatment in SH-SY5Y and LA-N-2 cells (Supplementary Figs. 3 and 4, respectively, Supplementary Data 2), 25–50 µM in HeLa cells (Fig. 1C, Supplementary Data 1), and 100 µM treatment in HEK 293 T, NIH 3T3, AML12 and LecCHO cells (Supplementary Figs. 5 to 8, Supplementary Data 2). Based on these observations, we chose the lower range treatment concentrations of 25 or 10 µM of MM-JH-2 for subsequent experiments in HeLa cells and in SH-SY5Y cells respectively.

### MM-JH-2 specifically labels gangliosides

Unnatural ManNAc derivatized sialic acid analogs can incorporate into lipids and proteins[24–29]. Following our previously proposed method for interrogating MCR selectivity[11], we examined the effect of cell permeabilization on MM-JH-2 fluorescence staining. Permeabilization of paraformaldehyde (PFA)-fixed HeLa cells with the nonionic detergent Triton X-100 resulted in the complete loss of MM-JH-2 fluorescence. Similar results were obtained by fixation and permeabilization with cold methanol. In contrast, the fluorescence signal was retained with saponin permeabilization. These results are consistent with MM-JH-2 labeling lipids since PFA fixes proteins but not lipids, Triton X-100 and cold methanol solubilize lipids, and saponin solubilizes only cholesterol (Supplementary Fig. 9A)[30,31]. To further establish that the labeling was associated with glycosphingolipids, lipids were isolated from DMSO and MM-JH-2-treated HeLa cells via Folch extraction[11]. Fluorescence was observed under UV illumination in the lipid extracts from the MM-JH-2 treated cells, supporting the idea that the compound was labeling lipids (Supplementary Fig. 9B). The lipid extracts from DMSO and MM-JH-2 treated HeLa cells were further analyzed by TLC, and a UV active spot was detected in the lipid extract from MM-JH-2 treated HeLa cells (Supplementary Fig. 9C). Furthermore, in-gel fluorescence scanning analysis of HeLa cell lysates treated with 50 µM or 100 µM MM-JH-2 for 72 hours, or with an equivalent volume of DMSO, showed no detectable protein labeling under UV illumination light (Supplementary Fig. 9D). Coomassie blue staining confirmed equal protein loading (Supplementary Fig. 9E), suggesting that MM-JH-2 does not label glycoproteins under these conditions.

We then sought to investigate the incorporation of MM-JH-2 in sialic acid-containing glycosphingolipids (gangliosides). Ganglioside biosynthesis begins with the synthesis of glucosylceramide (GlcCer) by glucosylceramide synthase (GCS), which in turn is converted to lactosylceramide (LacCer) (Fig. 2A). To investigate the incorporation of MM-JH-2 into gangliosides, ganglioside biosynthesis was disrupted by chemically inhibiting GCS. Selective inhibition of GCS by the inhibitor Genz-123346 blocks the synthesis of GlcCer, reducing glycosphingolipid biosynthesis[32]. Treatment with Genz-123346 at a concentration of 2 μM for 72 hours significantly reduced MM-JH-2 labeling. Surprisingly, at a higher concentration of inhibitor (5 μM), the MM-JH-2 signal intensity slightly increased (Supplementary Fig. 10, Supplementary Data 2).

The most abundant gangliosides formed in HeLa cells are descendants of GM3 gangliosides[33,34]. The very first ganglioside of the series, GM3, is encoded by the gene *ST3GAL5* (GM3 synthase). GM3 synthase is a Golgi-resident transferase that transfers the sialic acid moiety from CMP-sialic acid to LacCer, transforming LacCer into GM3 (Fig. 2A). GM3 serves as the precursor ganglioside for extension and further branching reactions, to produce the a-series gangliosides[3,35]. To confirm that MM-JH-2 labels gangliosides, *ST3GAL5* was knocked down in HeLa cells (Fig. 2C, western blot; Supplementary Fig. 19A for uncropped western blot, Supplementary Data 1) and SH-SY5Y cells (Supplementary Fig. 11B, western blot; Supplementary Fig. 19B for uncropped western blot) using siRNA silencing. Confocal imaging of these cells demonstrated concomitant loss of MM-JH-2 fluorescence (Fig. 2B and Supplementary Data 1 for HeLa cells, and Supplementary Fig. 11A and Supplementary Data 2 for SH-SY5Y cells), suggesting that MM-JH-2 is enzymatically labeling gangliosides.

Formation of MM-JH-2 modified gangliosides was confirmed by MALDI-TOF-MS analysis in NIH 3T3 cells which produce predominantly GM3 gangliosides[36] (Supplementary Fig. 12A-C). A lipid extract from NIH 3T3 cells was subjected to Folch extraction and the blue-fluorescent upper organic layer was analyzed by MALDI-TOF-MS. Distinct mass peaks for MM-JH-2 modified GM3 (GM3-MM-JH-2) ganglioside molecular species, arising from different ceramide moieties, were identified in the mass spectrum (Fig. 2D and Supplementary Fig. 12A–C). MALDI-TOF-MS analysis was also used to assess the abundance of the various endogenous ganglioside species, and revealed labeling of each of the major ganglioside species in the cell types studied (Supplementary Fig. 12D).

## MM-JH-2 colocalizes with the acidic components of cells at steady state

Having demonstrated metabolic incorporation of MM-JH-2 into gangliosides, the steady-state labeling pattern of the probe was examined in greater detail[37,38]. First, the biosynthetic capacity and colocalization with different cellular organelles was examined (Fig. 3A). HeLa cells treated with either DMSO or MM-JH-2 were separately co-stained with wheat germ agglutinin (WGA)-Texas red (a cell surface lectin and Golgi apparatus marker), ConA-Texas red (an endoplasmic reticulum (ER) marker), LAMP1 (a lysosomal marker), or LysoTracker deep red (a marker of lysosomes and late endosomes). Confocal imaging was employed to examine the extent of colocalization as determined by Pearson's R value (Fig. 3B, Supplementary Data 1). MM-JH-2 colocalized strongly with LAMP1 (mean Pearson's R value 0.67), and the colocalization was even greater with LysoTracker (mean Pearson's R value 0.79). On the other hand, colocalization with the ER (mean Pearson's R value 0.25), and plasma membrane (PM)/Golgi apparatus (mean Pearson's R value 0.17) were relatively lower (Fig. 3B, Supplementary Data 1). The lysosomal and late endosomal localization of MM-JH-2 is consistent with the established observation that bulky fluorophores in lipid residues, or biotin substitution at the *N*-acetyl position of sialic acid in gangliosides, restricts the segregation of gangliosides into the ordered lipid domains within the PM, and instead directs them towards acidic vesicles (lysosomes and late endosomes) within the cells[39].

Given the targeting of MM-JH-2 to acidic compartments during steady-state labeling, the influence of lysosomal function on MM-JH-2 labeling was assessed. Bafilomycin A1 is an inhibitor of vacuolar H$^+$ ATPase[40], effectively preventing acidification of lysosomes and late endosomes. As a result, it also reduces the activity of lysosomal enzymes which are optimally active at low pH[41]. With increasing concentration of Bafilomycin A1, the amount of MM-JH-2 fluorescence (normalized against red nuclear dye) increased significantly (Supplementary Fig. 13A, Supplementary Data 2). Complete loss of LysoTracker red signal upon treatment with 5 nM of Bafilomycin A1 confirmed inhibition of lysosomal acidification and hence the degradation capacity of lysosomal hydrolases and glycosidases (Supplementary Fig. 13B).

A pulse-chase labeling study was performed to track MM-JH-2 localization and trafficking using the intrinsic fluorescence of the probe. For this, HeLa cells were treated with MM-JH-2 for 15 minutes before replacing the MM-JH-2-supplemented media with fresh MM-JH-2-free media. Cells were imaged by confocal microscopy at different times post-labeling (Fig. 3C). Detectable fluorescence was observed within 15 minutes of MM-JH-2 treatment, with distribution consistent with labeling in the cytosol and Golgi apparatus. At 30 minutes, almost all the signal was associated with vesicle-like structures near the plasma membrane. After 1 to 2 hours fluorescence was largely localized to vesicular structures, consistent with endosomal internalization. Finally, lysosomal localization and catabolism of MM-JH-2 was detected with prolonged incubation (Fig. 3C). This observation of MM-JH-2 modified ganglioside salvaging *via* rapid endocytosis for lysosomal degradation is consistent with the established reports of [$^3$H]- or [$^{14}$C]-radiolabeled ganglioside internalization within 10-15 minutes, especially when the labeling is on the sialic acid residue[42].

Endocytosis of MM-JH-2 was further examined using chloroquine as a chemical inhibitor of endocytosis. Chloroquine is a well-known inhibitor of clathrin-dependent endocytosis that exerts its effects by blocking the function of clathrin and clathrin-coated vesicles[43]. eGFP-tagged caveolin and dynamin were also employed to track endocytic events. Another pulse-chase experiment was conducted wherein cells were treated with chloroquine-supplemented or control media prior to MM-JH-2 labeling. When imaged after 4 hours, a considerable fraction of the MM-JH-2 fluorescence remained membrane-associated in chloroquine treated cells compared to the predominantly vesicular distribution that colocalized with LysoTracker fluorescence in untreated cells (Supplementary Fig. 14A). To define the route of entry, HeLa cells were transfected separately with eGFP-tagged WT dynamin or WT caveolin, followed by treatment with MM-JH-2 or DMSO. The extent of the colocalization was measured by Pearson's R value (Supplementary Fig. 14B and C). Both eGFP-WT dynamin and eGFP-WT caveolin were considerably colocalized with MM-JH-2 (mean Pearson's R value 0.44 for WT dynamin and 0.36 for WT caveolin, Supplementary Fig. 14C, Supplementary Data 2), indicating that endocytosis of MM-JH-2 may involve both dynamin and caveolin. Endocytosis was next perturbed using the K44A mutant of dynamin to block endocytosis. In an experiment where the eGFP-dynamin K44A mutant was used, MM-JH-2 colocalized almost exclusively with eGFP signals (mean Pearson's R value 0.78, Supplementary Fig. 14C). The absence of any fluorescence upon 405 nm excitation in the DMSO-treated sample eliminates the possibility of fluorescent signal interference between eGFP and the blue-fluorescent signal (Supplementary Fig. 14B). Together, these experiments indicate that MM-JH-2 endocytosis is dependent on both dynamin and caveolin, as previously suggested for gangliosides[44]. The effects of the dominant negative K44A mutant blocking endocytosis suggest that all the labeled ganglioside species build up in endocytic structures when internalization is blocked.

MM-JH-2 labeling was also compared with the previously reported terminal alkyne-ManNAc analog Pr$_4$ManNAlk[37]. HeLa cells were treated with either MM-JH-2, Pr$_4$ManNAlk, or both for 72 hours. Clickable Pr$_4$ManNAlk was labeled with AlexaFluor 488 azide *via* a CuAAC reaction, and detected using 488 nm laser excitation, whereas MM-JH-2 labeling was visualized using 405 nm laser excitation. The cell surface lectin and *N*-acetyl glucosamine marker WGA was detected using 560 nm laser excitation. MM-JH-2 labeling was strongly and reproducibly present in intracellular structures, but not strongly localized to the cell surface. In contrast, the promiscuous MOE reagent Pr$_4$ManNAlk was distributed predominantly on

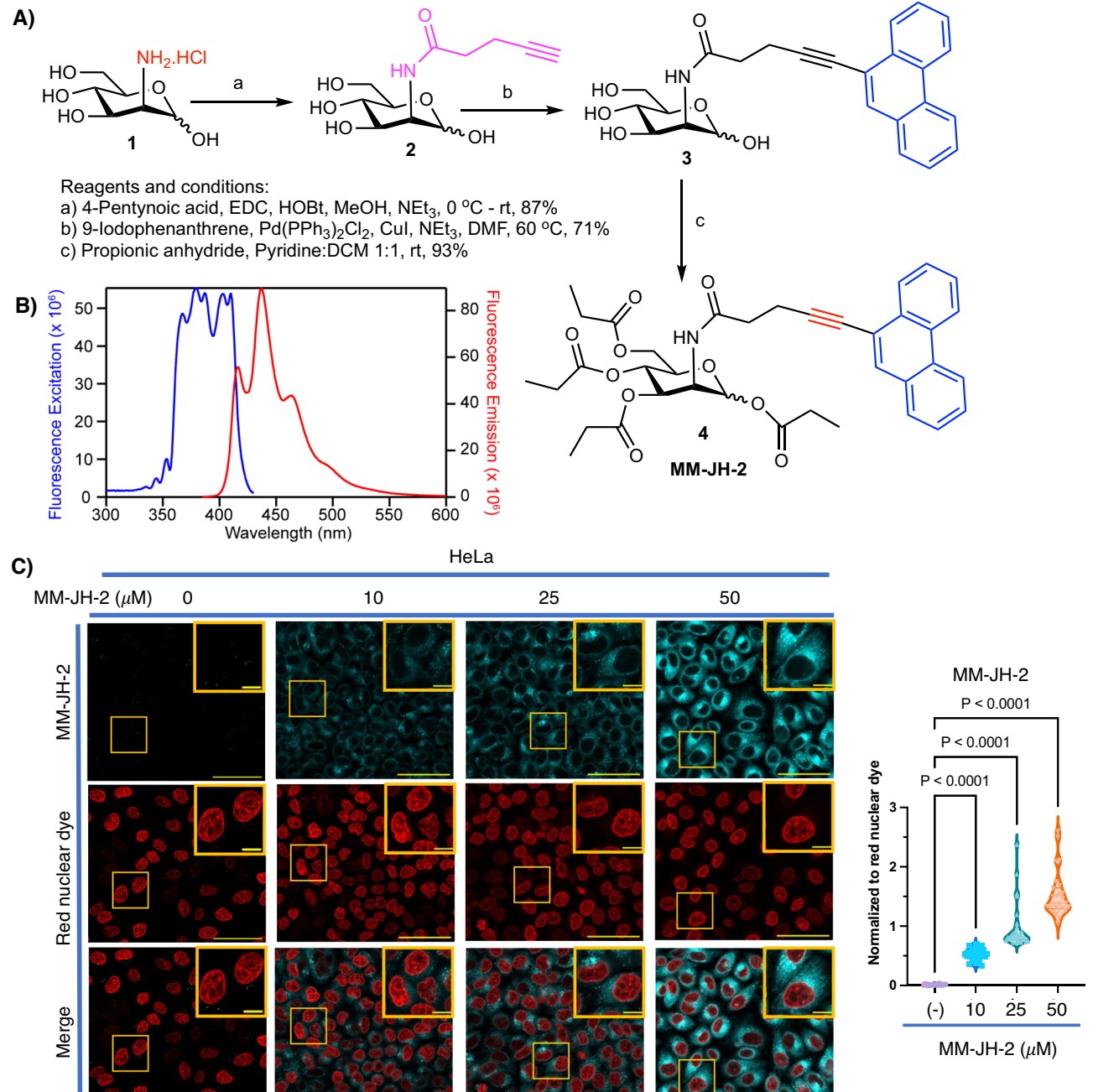

**Fig. 1 | Synthesis and fluorescent labeling of cells with MM-JH-2. A** Chemical synthesis of the title compound MM-JH-2. **B** Fluorescence excitation (blue) and emission (red) spectra of MM-JH-2. **C** MM-JH-2 was incorporated into HeLa cells in a concentration-dependent manner. $N = 5$ individual biological replicates, $n = 26$ individual cells chosen for quantification from the confocal images. An ordinary one-way ANOVA test was performed. *P*-values are shown in the graph, and error bars represent the standard deviation centered on the mean. Scale bars are 50 µm and 10 µm for zoomed images. Quantification is shown to the right of the images.

the PM with some nonspecific labeling of the nucleolus following click chemistry (Supplementary Fig. 15A). In cells treated with both MM-JH-2 and Pr₄ManNAlk, juxtaposition of blue and green fluorescence indicated the signals coming from the labeling by these two different ManNAlk derivatives colocalized in intracellular structures but not on the PM nor in the nucleus.

Using ganglioside-specific labeling by MM-JH-2 as a positive control, the detergent solubility of lipids was exploited to determine the fraction of Pr₄ManNAlk labeling attributable to PFA-fixable proteins versus non-fixable lipids (Supplementary Fig. 15B, Supplementary Data 2). HeLa cells were treated with 100 µM of Pr₄ManNAlk and 25 µM of MM-JH-2, and then incubated for 72 hours. Cells were then fixed with PFA, followed by

Triton X-100 permeabilization and labeling of the Pr₄ManNAlk terminal alkyne *via* click reaction with AlexaFluor 488 azide. Triton X-100 permeabilization resulted in the complete loss of detergent-soluble MM-JH-2 fluorescence. In comparison, the Pr₄ManNAlk signal was reduced by ~10%, suggesting that only a small fraction of glycan labeling is due to labeling of gangliosides by the promiscuous MOE reagent Pr₄ManNAlk.

## Raman spectral imaging correlates MM-JH-2 and lipid distributions

In addition to the fluorescent properties of MM-JH-2, the alkyne moiety produces a unique vibrational signature in the quiet region of the cellular Raman spectrum, where there are no endogenous biomacromolecular

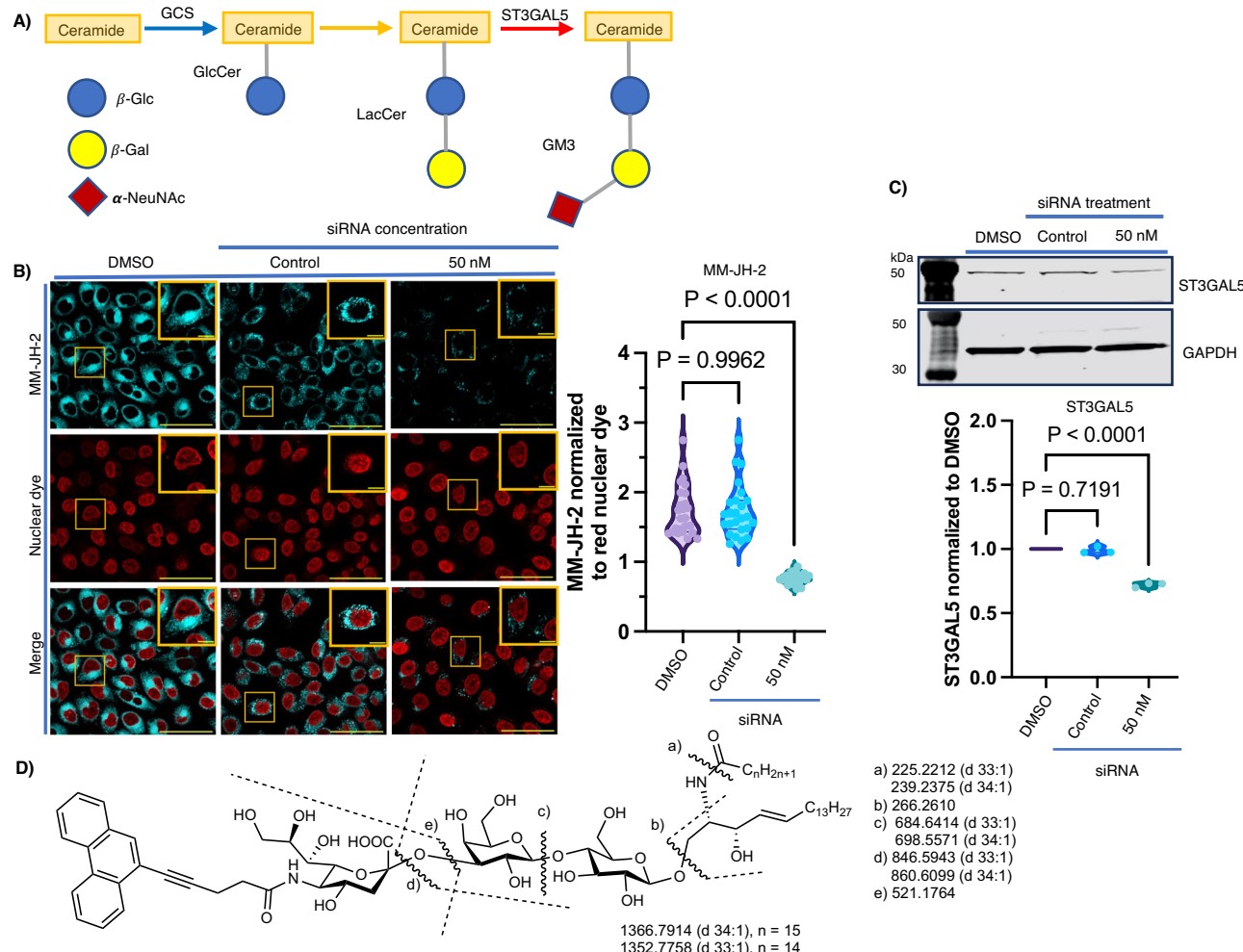

**Fig. 2 | MM-JH-2 enzymatically labels gangliosides. A** Schematic representation of GM3 ganglioside biosynthesis. **B** *ST3GAL5* siRNA knockdown reduces MM-JH-2 labeling in HeLa cells as visualized by confocal fluorescence imaging. $N = 5$ individual biological replicates, $n = 25$ individual cells chosen for quantification from the confocal images. An ordinary one-way ANOVA test was performed. *P*-values are shown in the graph and error bars represent the standard deviation centered on the mean. Scale bars are 50 μm and 10 μm for zoomed images. Quantification is shown to the right of the images. **C** Western blots showing that siRNA against *ST3GAL5* in HeLa cells reduces *ST3GAL5* levels by about 40%. An ordinary one-way ANOVA test was performed. $N = 3$ individual biological replicates. *P*-values are shown in the graph and error bars represent the standard deviation centered on the mean. Quantification is shown below the images. **D** MALDI mass fragments identified from ganglioside extracts of MM-JH-2 treated NIH 3T3 cells.

vibrational modes[18,45]. This C≡C stretching band occurs between 2200 and 2245 cm$^{-1}$, centered at ~2225 cm$^{-1}$ (Fig. 4A). This band is clearly detectable in HeLa cells treated with MM-JH-2 (Fig. 4B, Supplementary Fig. 16). Although the alkyne stretching band overlaps with the water bend-libration combination band (1900 to 2500 cm$^{-1}$, recently demonstrated as a sensitive reporter of water structure in cells[46]), the sharp alkyne peak is clearly resolved compared to the relatively low intensity and broad bend-libration band. Despite the presence of the water bend-libration band, mapping the C≡C stretching spectral region in DMSO-treated (control) HeLa cells produces only noise (Supplementary Fig. 16). Thus, the alkyne stretching mode provides an unambiguous identifier of the compound in the complex cellular milieu.

To map the spatial distribution of MM-JH-2 in treated HeLa cells, we first identified cells of interest for Raman spectral imaging by bright-field and widefield epifluorescence imaging using the intrinsic fluorescence of the probe (Fig. 4C). Raman maps of MM-JH-2, lipids, and nucleotides were generated by integrating the area of the alkyne stretching band (2200–2245 cm$^{-1}$), the methylene C–H stretching band (2835–2856 cm$^{-1}$, primarily arising from lipids) and the symmetric O–P–O stretch of the polynucleotide phosphodiester backbone (763–794 cm$^{-1}$). First, a clear

overlap between the epifluorescence image and alkyne map is apparent. Second, comparison of the alkyne, lipid, and nucleotide maps indicates localization of MM-JH-2 to the cytoplasm and exclusion from the nucleus (Fig. 4D). Although the distribution of the alkyne closely matches that of lipids, it appears to be absent from free lipid droplets (distinguishable as intense puncta in the lipid maps) relative to other structures.

### MM-JH-2 preferentially labels cells with elevated ganglioside flux

To gain further insight into the incorporation of MM-JH-2, the probe was used to treat different cell types which are known to differ in the composition, distribution, and location of gangliosides[47]. We assessed the cell-type-specific labeling of gangliosides by MM-JH-2. MM-JH-2 labeling was clearly detectable in MCF7 malignant breast cancer cells at a treatment concentration of just 5 μM (Fig. 5A, Supplementary Data 1). Conversely, the nonmalignant breast cancer cells, MCF10A did not exhibit detectable MM-JH-2 labeling until about 100 μM treatment (Fig. 5B, Supplementary Data 1). This augmented incorporation of MM-JH-2 in malignant cells compared to nonmalignant cell lines is indicative of high levels of gangliosides in the malignant cells. This cell-type-specific labeling of MM-JH-2 is consistent with reports that the lipid and sialic acid content of cervical or

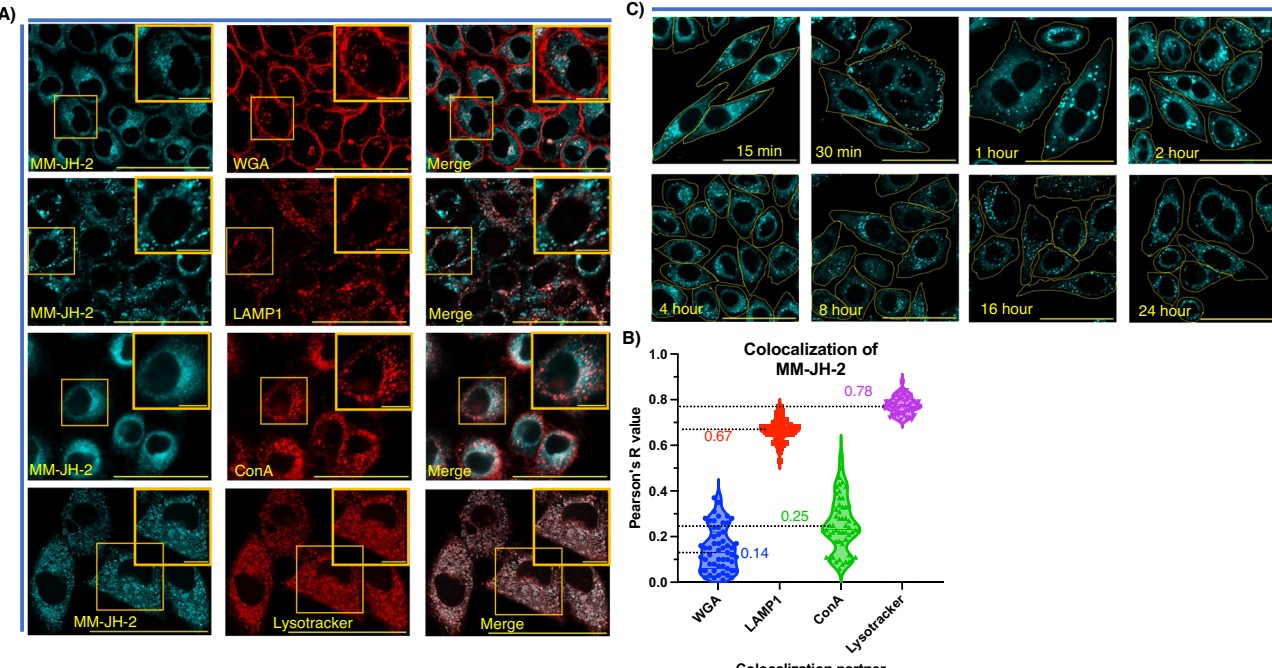

**Fig. 3 | MM-JH-2 localizes to lysosomes and late endosomes for catabolism.**
**A** Confocal fluorescence images showing colocalization of MM-JH-2 in blue with
WGA (plasma membrane and Golgi), LAMP1 (lysosomes), ConA (ER) and Lyso-
Tracker (lysosomes and late endosomes) in red. Colocalization was quantified by
Pearson's R value as indicated to the right of the images. $N = 3$ individual biological
replicates, $n = 55$ individual cells chosen for Pearson's R value calculation from the
confocal images. Scale bars are 50 μm and 10 μm for zoomed images. **B** Mean
Pearson's R value calculation for the confocal images. The error bars represent
standard deviation centered on the mean. **C** Confocal fluorescence images showing
time-dependent incubation of HeLa cells with MM-JH-2 show rapid endocytosis
and turnover of the label. $N = 3$ individual biological replicates, Scale bars are 50 μm.

mildly malignant breast cancer cell lines (HeLa or MCF7) is significantly
higher than MCF10A cells[48].

Considering this observation, several malignant and non-malignant
cell lines were assessed for differential effects of MM-JH-2 treatment. SH-
SY5Y, LA-N-2, HepG2, MCF7, MCF10A, HeLa, and female MEF cells were
treated with increasing concentrations (0 to 500 μM) of MM-JH-2 for
72 hours, and cytotoxicity was then assessed using MTT assays (Supple-
mentary Fig. 17A and B, Supplementary Data 2). The IC$_{50}$ of MM-JH-2
in neuroblastoma cell lines (SH-SY5Y and LA-N-2) was relatively low
(~21 μM for SH-SY5Y and ~34 μM for LA-N-2, respectively). The IC$_{50}$ of
MM-JH-2 was relatively moderate in cervical and breast cancer lines (~67
μM for HeLa and ~74 μM for MCF7, respectively), and relatively high in the
hepatocarcinoma cell line HepG2 (~122 μM). In contrast, MM-JH-2 did not
demonstrate appreciable cytotoxicity in the nonmalignant MEF cells or
MCF10A. The much higher IC$_{50}$ value for the phenanthrene-lacking
compound Pr$_4$ManNAlk in HeLa cells (~392 μM), suggests that the cyto-
toxic effect of MM-JH-2 can be ascribed to the inclusion of the phenan-
threne group on the terminal alkyne. ManNAc treatment was also not
associated with notable cytotoxicity in HeLa cells.

### MM-JH-2 differentially labels ex vivo-treated mouse T and B cells
To assess the utility of MM-JH-2 as an endogenous labeling reagent, the
probe was tested for differential staining of ex vivo immune cell types. T and
B cells were selected for this experiment. These cells are derived from a
common progenitor, have similar physical features such as size and shape,
and both contribute to adaptive immunity[49,50]. Despite these similarities,
these cell types perform unique functions, with B cells generating antibodies
to neutralize invading pathogens, and T cells recognizing receptor-bound
antigens and acting as effector cells for cell-mediated immunity[49]. Generally,
cell surface antibody markers are used to distinguish T and B cells, but
differences in lipid profiles have recently been exploited to differentiate these

cell types. The Lipid-Oriented Live-cell Distinction (LOLD) technique has
successfully distinguished B and T lymphocytes. B cells were found to
maintain higher flexibility in the cell membrane than T cells, and pre-
ferentially accumulate lipid-like probes[51,52]. In addition, *ST3GAL5* gene
expression in B-cells is twice as high as in T-cells. (http://biogps.org/#goto=
genereport&id=8869). These properties make T and B cells an ideal model
system for ex vivo staining by MM-JH-2.

B and T cells were collected from C57/BL6 mouse spleens[53]. The
harvested lymphocytes were incubated with different concentrations of
MM-JH-2 over varying treatment times. Flow cytometry analysis of the
splenocytes was performed with antibodies against T-cell markers (CD4,
CD8), and the B-cell marker B220. MM-JH-2 fluorescence was used to
determine the extent of labeling in each cell population. MM-JH-2 positive
T and B lymphocytes were detectable at a treatment concentration of 50 μM,
and mean fluorescent intensity (MFI) increased with increasing MM-JH-2
concentration (Fig. 6A–E, Supplementary Data 1). Both MFI and the per-
centage of MM-JH-2 positive cells were significantly greater in B cells
(B220$^+$) than T cells (CD4$^+$ or CD8$^+$), irrespective of incubation time
(2 hours or 4 hours) (Fig. 6A–E, Supplementary Data 1, Supplementary
Fig. 18). These findings confirm a previous report using LOLD, and establish
the utility of MM-JH-2 for ex vivo labeling of live animal cells.

### Discussion
This article describes the synthesis and characterization of MM-JH-2 as a
dual Raman-fluorescent probe for the selective labeling of gangliosides. This
probe has utility in the interrogation of ganglioside biosynthesis, and has
potential as a diagnostic and therapeutic agent. The motivation behind
design and synthesis of MM-JH-2 was: (i) the phenanthrene ring in con-
jugation with the alkyne produces a detectable fluorescence signal that was
used to determine its subcellular distribution, (ii) the unequivocal Raman
fingerprint of the non-terminal alkyne signal enables detection of the

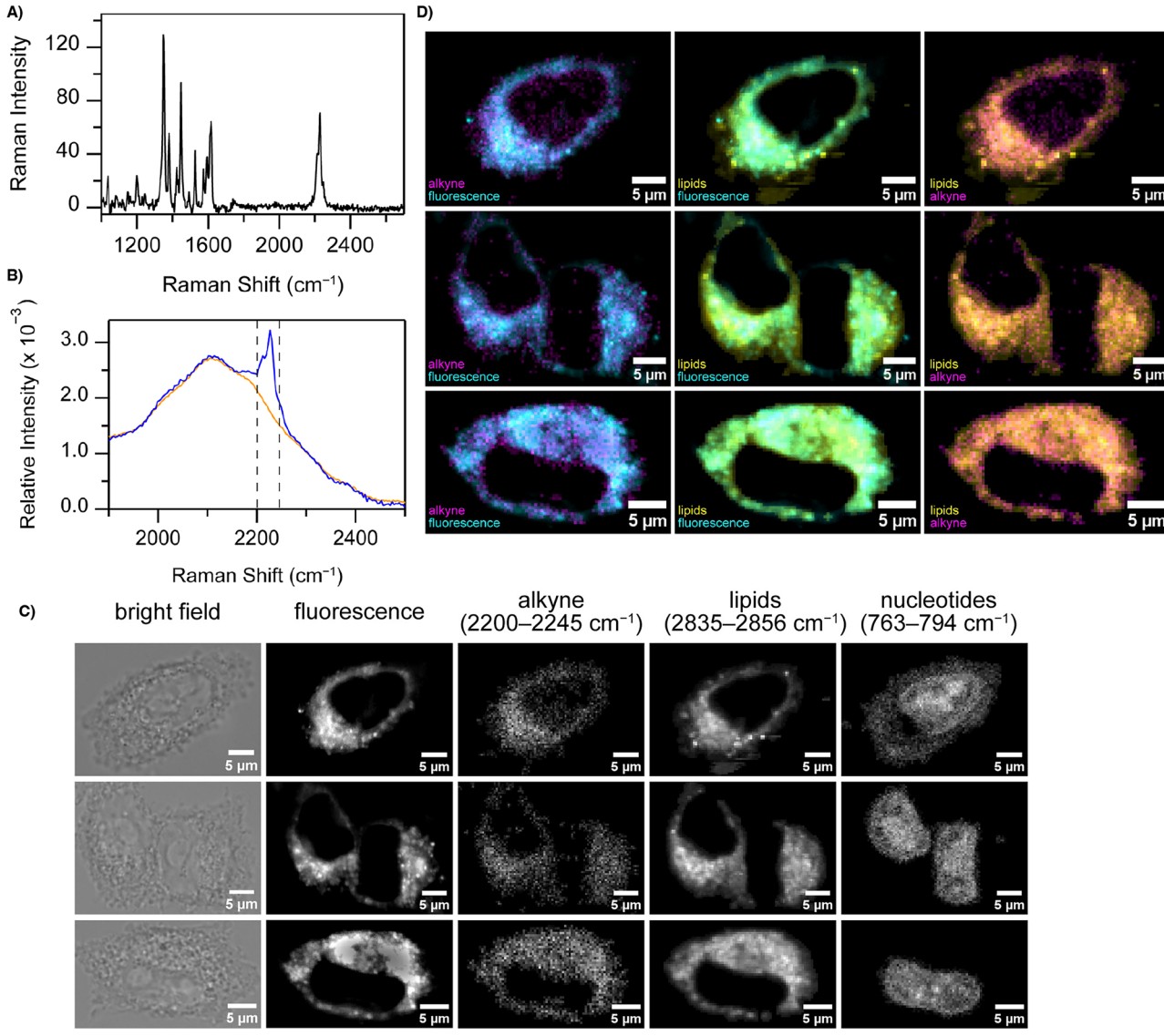

**Fig. 4 | Raman spectral imaging of MM-JH-2 treated HeLa cells. A** Spontaneous Raman spectrum of MM-JH-2 in DMSO. **B** Comparison of Raman spectral region of 1900–2500 cm$^{-1}$ showing averaged whole-cell spectra of control HeLa cells (DMSO, orange) and Hela cells treated with 25 μM MM-JH-2 (blue). Dashed lines indicate the spectral region integrated to generate Raman maps of MM-JH-2 distribution. Full spectra are shown in Supplementary Fig. 16. **C** Corresponding bright-field images, widefield epifluorescence images, and Raman maps (spectral ranges as indicated) of HeLa cells treated with MM-JH-2. **D** Merged images of the MM-JH-2 alkyne stretching band map (magenta), lipid C–H stretching band map (yellow) and MM-JH-2 fluorescence (cyan).

chemical environment of the probe along with simultaneous label-free imaging of cellular components, and (iii) hydrophobicity of the bulky phenanthrene ring directed the MOE towards lipids rather than proteins.

Multiple cell lines separately treated with MM-JH-2 metabolized the bioorthogonal sugar and accumulated the fluorophore in a concentration-dependent manner. MM-JH-2 fluorescence disappeared upon treatment with the lipid-dissolving detergent Triton X-100, but survived the β-hydroxy sterol-acting reagent saponin, indicating that MM-JH-2 does not label sialic acid-containing glycoproteins, but rather glycolipids, i.e., gangliosides. Blue fluorescence of the whole and TLC-analyzed lipid extracts from HeLa cells treated with MM-JH-2 further confirmed that the probe acts as a lipid labeling probe. In-gel fluorescence scanning analysis of HeLa cell lysates treated with MM-JH-2 showed no detectable protein labeling under UV illumination. Raman spectral imaging was used to demonstrate localization of the probe in the lipid-rich cytoplasm. The value of the alkyne functional group is apparent when comparing of the clearly detectable C≡C stretching band of MM-JH-2 in cells to other characteristic

vibrational features of the probe, which, though detectable, overlap considerably with vibrational bands of endogenous biomolecules (Supplementary Fig. 16).

Within 15 minutes of treatment, time-dependent labeling experiments showed MM-JH-2 fluorescence in structures consistent with the known biosynthetic pathway of gangliosides. The probe showed transport to the PM, with a very short residence time at the PM, before being rapidly endocytosed to late endosome and lysosome-like acidic organelles. MM-JH-2 fluorescence intensity was directly correlated with the acidity and, hence the degradation capacity of those organelles. Blocking acidification of these structures prevented degradation of the probe and led to an accumulation of the probe in late endosomes/lysosomes. Genetic perturbation of ganglioside biosynthesis *via* siRNA knockdown of the gene encoded *ST3GAL5* (GM3 synthase) reduced the MM-JH-2 labeling in multiple cell lines. Finally, MALDI-TOF mass spectral analysis of the lipid extracts directly confirmed the presence of MM-JH-2 modified gangliosides. Collectively, these results demonstrate that MM-JH-2 is the first ganglioside-specific MOE allowing dual detection by Raman and fluorescence microscopy.

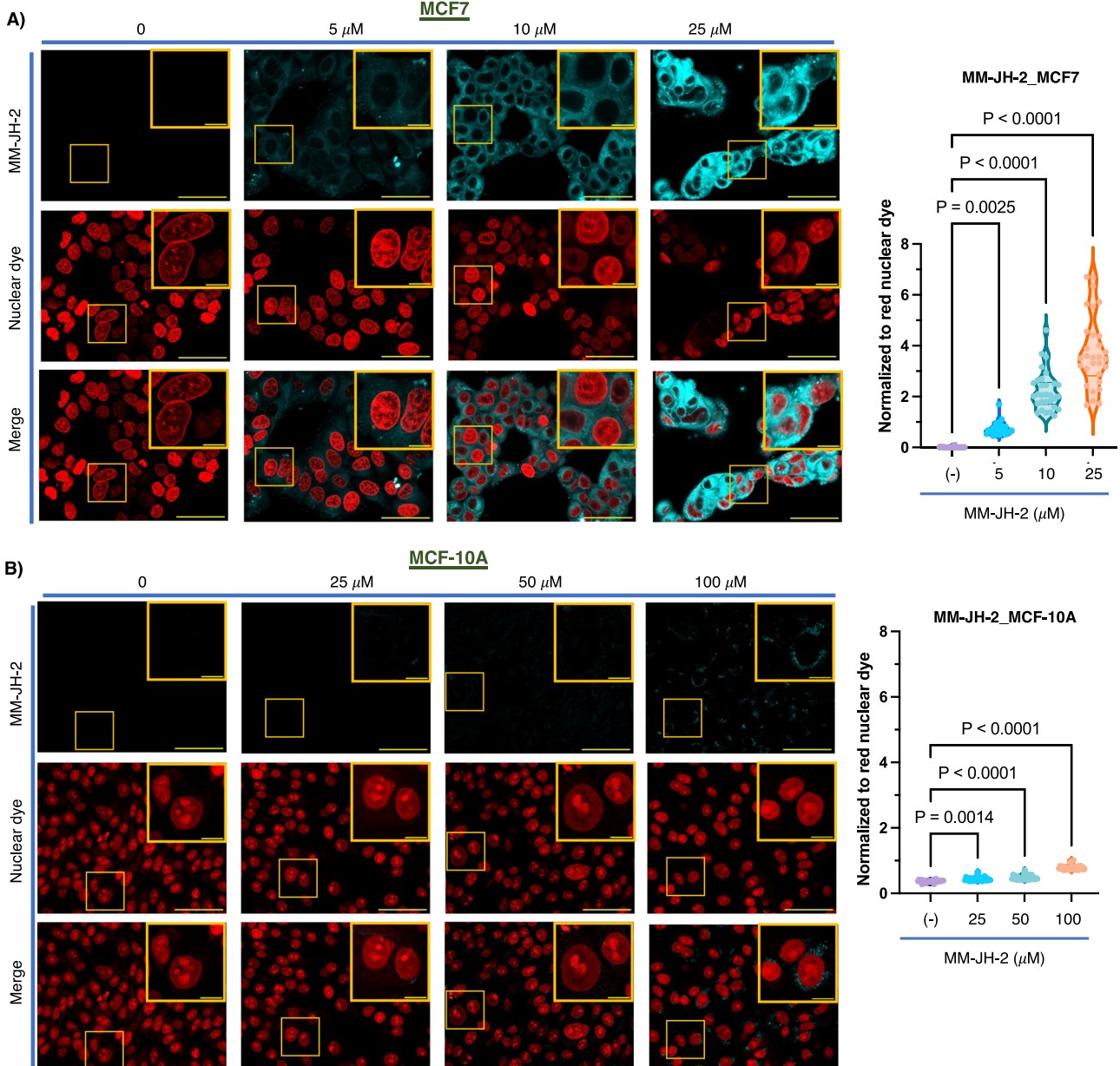

**Fig. 5 | MM-JH-2 selectively detects cancer cells over nonmalignant cells. A** MM-JH-2 labels MCF7 cells in a concentration dependent manner with a detectable signal at 5 μM treatment. $N = 5$ individual biological replicates, $n = 30$ individual cells chosen for quantification from the confocal images. **B** MM-JH-2 only weakly labels nonmalignant MCF10A cells even at 100 μM treatment. $N = 5$ individual biological replicates, $n = 26$ individual cells chosen for quantification from the confocal images. An ordinary one-way ANOVA test was performed. *P*-values are shown in the graph and error bars represent the standard deviation centered on the mean. Scale bars are 50 μm and 10 μm for zoomed images. Quantification is shown to the right of the images.

## MM-JH-2, a selective probe for monitoring ganglioside biosynthesis

Cancer cells known to have high ganglioside content, including HeLa, MCF7, LA-N-2 and SH-SY 5Y, were strongly labeled at very low concentrations (10 μM) of MM-JH-2, whereas nonmalignant MCF10A cells had nearly inconspicuous levels of MM-JH-2 incorporation under the same conditions. Treatment with MM-JH-2 specifically detected gangliosides and selectively killed cancer cells in a concentration-dependent manner. We hypothesize that this cancer-cell-specific cytotoxicity can be attributed to the abnormally elevated ganglioside expression levels that are a hallmark of cancer cells. MM-JH-2 was also useful as a probe for differentiating biosynthetic flux in components of the circulating immune system. Utilizing the differences in lipid concentration between lymphocytic B and T cells, MM-JH-2 treatment was able to differentiate B220[+] and CD4[+] or CD8[+]

cells, thus providing evidence that this compound can be incorporated into cells, or possibly tissue biopsies, for diagnostic or therapeutic purposes.

Within the cell, synthesis of gangliosides begins in the ER, with elongation of the glycan moiety occurring in the Golgi apparatus by sequential addition of different carbohydrate molecules to the existing LacCer acceptor molecule. The probe we have developed would be expected to be incorporated in the Golgi complex, followed by transport to the PM *via* vesicles with exocytic membrane flow largely governed by the gradient of the hydrophobicity of the molecule[42]. The ganglioside molecule is comprised of the hydrophilic saccharide portions that are exposed to the cell surface and the hydrophobic ceramide moieties that are firmly embedded into the cytosolic leaflet of the lipid membrane[1,42]. Following residence on the PM, gangliosides are internalized *via* endocytosis and degraded in acidic compartments (late endosomes and lysosomes), regenerating sugar and lipid fragments to

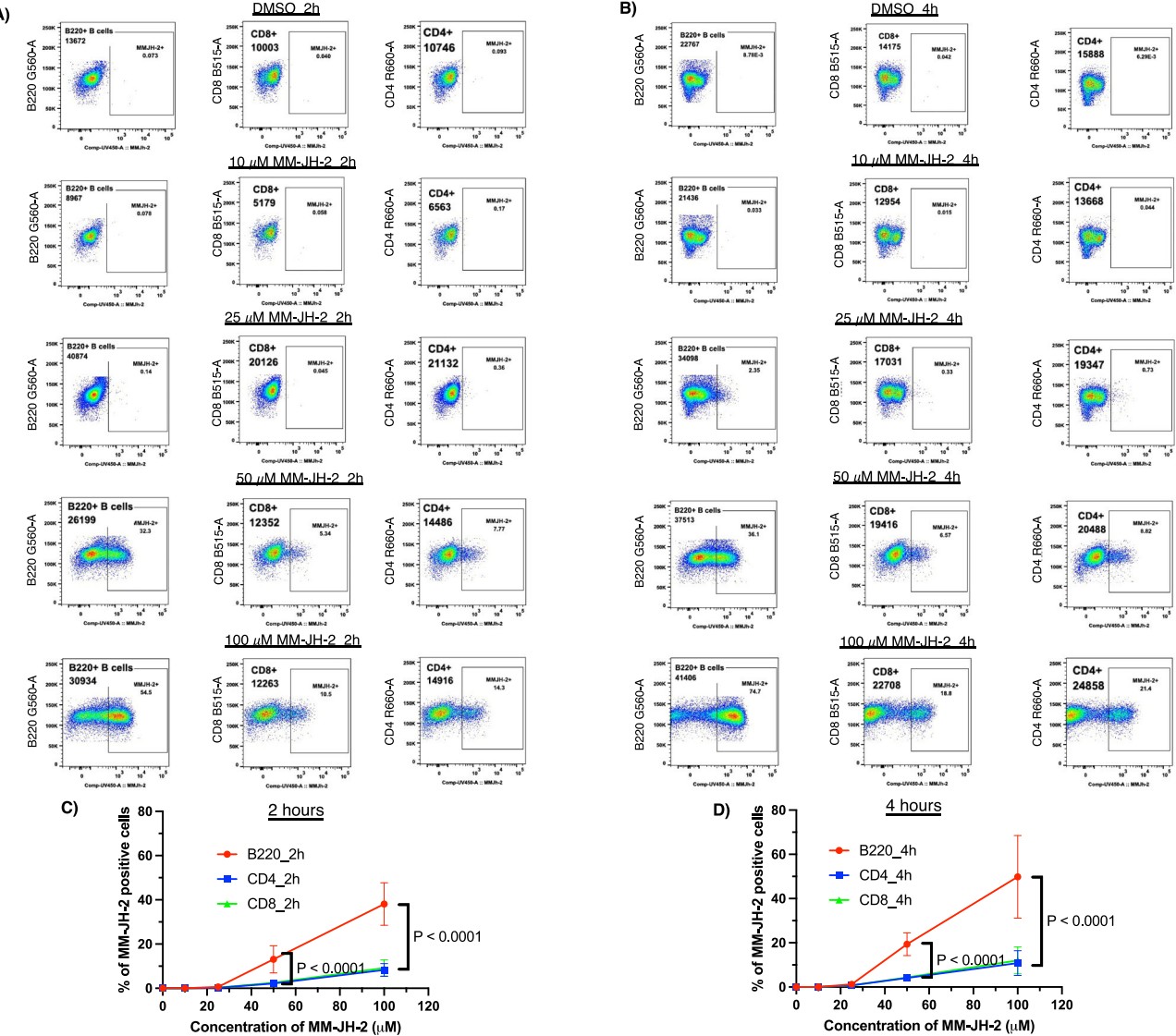

**Fig. 6 | MM-JH-2 preferentially labels B cells over T cells in a coculture of splenocytes.** Histograms of splenocyte cell populations after **A** 2 hours and **B** 4 hours of treatment with MM-JH-2. **C** Change in the percentage of MM-JH-2 positive cells over 2 hours of treatment. **D** Change in the percentage of MM-JH-2 positive cells over 4 hours of treatment. $N = 6$ individual biological replicates. An ordinary two-way ANOVA test was performed. *P*-values are shown in the graph, and error bars represent the standard deviation centered on the mean.

complete the glycan recycling process. Multiple attempts to decipher the precise half-life of ganglioside turnover have been reported, but due to salvage pathways, the reported numbers (from 15 minutes to 3 days depending on the cells or methods used for detection) are unreliable and likely higher than the actual values[42,54]. Inherent structural amphiphilicity, along with uncertain biosynthesis rates and lack of proper tools to study gangliosides in real time, have long complicated this aspect of glycobiology. The data presented here suggest that MM-JH-2 may have promise as a tool to elucidate further details of this process.

To prove that the large hydrophobic phenanthrene substitution preferentially directed the unnatural monosaccharide towards lipids, PFA-fixed and Triton X-100-permeabilized HeLa cells were imaged and compared with PFA-fixed and saponin-permeabilized cells. The non-ionic detergent Triton X-100 interacts with unfixable lipids and amphipathic membranes, dissolving the glycosphingolipids[55]. Saponin dissolves only cholesterols present in the membrane, and aids in visualization of cholesterol-rich intracellular organelles (thus excluding the nucleus and mitochondria)[30]. The post-fixation loss of MM-JH-2 fluorescence upon Triton X-100

permeabilization, but retention of the fluorescence upon saponin permeabilization, indicates that MM-JH-2 labeling is not associated with proteins, but instead with a non-fixable cellular component, glycosphingolipids. Whereas sialic acid and GlcNAc were detected by WGA in the PM and Golgi apparatus of non-permeabilized cells, in saponin-permeabilized cells, they were detected only in the Golgi, and in Triton X-100-permeabilized cells, they were detected in both the Golgi and nucleus.

Functional group modifications on the N-acetyl group of the ManNAc molecule are well tolerated by the sialic acid biosynthetic pathway, resulting in the formation of their corresponding CMP-sialic acid derivatives[37]. Sialyltransferase (ST) selectivity is influenced by multiple structural and biochemical factors. Key determinants include conserved protein motifs, the architecture of the catalytic site, and the length and flexibility of the ST stem region, which impacts substrate accessibility[56]. The chemical and spatial properties of acceptor substrates further modulate enzyme specificity[56]. Ligand-induced conformational shifts can alter ST activity and substrate affinity[56]. Additionally, STs form homo- and heterooligomeric complexes that facilitate substrate channeling and enhance donor and acceptor

recognition[56]. Their precise localization within cellular organelles and incorporation into supramolecular enzyme assemblies regulate substrate accessibility and catalytic efficiency. Sequence alignments of the catalytic domain identify highly conserved sialylmotifs—L (long), S (short), III, and VS (very short). Residues within motifs L and S mediate donor and acceptor substrate binding, whereas motifs III and VS contain catalytic residues critical for enzymatic activity[57,58].

Our interpretation of our current results is that the hydrophobic nature of the bulky phenanthrene ring directs the MOE probe preferentially toward lipid targets rather than proteins. The lipid species that could act as acceptors are the ceramide-anchored gangliosides. Moreover, the steric bulk of the CMP-phenanthrene-9-SiaAlk derivative appears to restrict its recognition to the sialylmotifs L and S of *ST3GAL5*, enzymes specifically involved in glycolipid biosynthesis. This observed selective incorporation into gangliosides serves to highlight important features of the largely structurally uncharacterized ganglioside-specific STs like *ST3GAL5*.

## MM-JH-2 as a tool for the pathophysiological role of gangliosides

Because tools to specifically examine gangliosides have long been lacking, much remains unknown regarding the biosynthesis of gangliosides and how changes in ganglioside content contribute to diseases like cancer. Gangliosides appear to behave as a double-edged sword in cancer, as they have been determined to be either pro- or anti-cancerous depending on the affected cell or tissue of interest. Gangliosides play an anti-tumoral role by inhibiting cell proliferation in gliomas, epidermoid carcinoma, neuroblastoma (NB), and astrocytoma, but they promote neuronal differentiation in PC12, and breast cancer cell lines 4T1 and 67NR, leading to enhanced tumorigenesis[59]. Patients carrying a rare mutation in the *ST3GAL5* gene develop infantile onset symptomatic epilepsy syndrome (West syndrome), dyspigmentation of the skin, and abnormal auditory responses[2]. Evidence from human and mouse studies strongly suggests that a decrease in endogenous gangliosides plays a key role in the pathogenesis of PD, Huntington's Disease (HD), depression, and anxiety. Although it remains unknown why this pathway becomes affected, and whether cerebro-spinal fluid (CSF) levels of gangliosides might be used as biomarkers of disease onset and/or progression. Abnormal ganglioside metabolism is also attributed to several primary and secondary lysosomal storage-related disorders like Sandhoff disease, Gaucher disease, Neiman-Pick disease, Farber disease, as well as Hurler, Hunter, Sanfilippo, and Sly syndrome[60]. The MOE we developed here will facilitate further understanding of the basic biology of gangliosides, the "factotum of nature"[61], and how we can mechanistically exploit ganglioside biosynthesis for both diagnostic and therapeutic purposes. Further studies of the effects of MM-JH-2 in more complex ganglioside systems, as well as application of the probe for diagnostic and therapeutic purposes, are ongoing and will be forthcoming.

## Materials and methods
### Reagents
All chemicals, reagents, and general lab supplies were purchased from Thermo Fisher Scientific or Sigma-Aldrich unless otherwise noted. Primary antibodies used include those for the following epitopes (catalog number): Mouse anti GAPDH (Abcam ab8245), Rabbit anti GAPDH (Abcam, ab18078), Rabbit anti ST3GAL5 (Novus Biologicals, NBP2-20492), Rabbit anti LAMP1 (Cell Signaling Technologies, 9091S), Allophycocyanin (APC)-CD4$^+$ (Thermo Fisher Scientific, 17-0042-82), Fluorescein isothiocyanate (FTIC)-CD8$^+$ (Thermo Fisher Scientific, 11-0081-85), Phycoerythrin (PE)-B220$^+$ (Thermo Fisher Scientific, 12-0452-82).

Inhibitors were purchased from different vendors with the following specifications: Bafilomycin-A1 (Cell Signaling, 54645), GenZ-123346 (Sigma-Aldrich, 5382850001), Chloroquine (Sigma-Aldrich, C6628).

Primary antibodies were used at a 1:1000 dilution in Odyssey PBS blocking buffer with 0.1% Tween 20 (PBSBBTw) for immunoblotting and at a 1:100 dilution in 0.1% BSA solution in PBS for immunofluorescence. Secondary antibodies included Odyssey IRDye 680 CW goat anti-mouse (Li-COR, 926-68070), IRDye 800 CW goat anti-mouse (Li-COR, 926-

32210), IRDye 680 CW goat anti-rabbit (Li-COR, 926-68071) and IRDye 800 CW goat anti-rabbit (Li-COR, 926-32211), and they were used at a 1:10,000 dilution in Odyssey PBSBBTw for immunoblotting. AlexaFluor 568 goat anti-rabbit (Invitrogen, A11011) was used at a 1:500 dilution in 0.1% BSA solution in PBS for immunofluorescence.

### General methods for chemical synthesis
Unless otherwise specified, all reagents and solvents were purchased from Thermo Fisher Scientific or Sigma-Aldrich Chemical Company and were used as supplied. Reactions were monitored by thin-layer chromatography (TLC) on silica gel 60 glass slides. Spots were visualized by charring with $H_2SO_4$ in EtOH (5% v/v) and/or UV light. Solutions in organic solvents were dried with anhydrous $MgSO_4$ and concentrated at reduced pressure at <40 °C.

### Chemical analysis instruments
NMR spectra were measured at 25 °C for solutions in CDCl$_3$, at 600 MHz for $^1$H, and at 150 MHz for $^{13}$C with Bruker Avance Spectrometers. Assignments of NMR signals were aided by 1D and 2D experiments ($^1$H–$^1$H homonuclear decoupling, APT, COSY, and HSQC) run with the TopSpin software supplied with the spectrometer. Chemical shifts were referenced to the signals of residual non-deuterated solvent for $^1$H (7.16 ppm), and the signal of the solvent for $^{13}$C (CHCl$_3$, 77.00 ppm). Topspin (v12.0.3) was used for all chemical NMR analysis. Data are reported as chemical shift, multiplicity (brs, broad signal; s, singlet; d, doublet; t, triplet; q, quartet; m, multiplet), coupling constants in Hertz (Hz), and integration area. High-resolution mass spectrometric (HRMS) measurements were performed on a proteomics optimized Q-TOF-2 (Micromass-Waters) using external calibration with polyalanine, unless otherwise noted. Observed mass accuracies are those expected based on known instrument performance, as well as trends in masses of standard compounds observed at intervals during the series of measurements. Reported masses are observed masses uncorrected for this time-dependent drift in mass accuracy.

### Synthesis
2-Deoxy-2-*N*-phenanthrene-9-pentynoylamide-1,3,4,6-tetra-*O*-propio-nyl-D-mannopyranoside (phenanthrene-9-Pr$_4$ManNAlk; MM-JH-2; **4**):

4-Pentynoic acid (341 mg, 3.48 mmol) was added to a stirring solution of D-mannosamine hydrochloride (**1**, 500 mg, 2.32 mmol) in triethylamine (TEA, 2 mL) and methanol (18 mL) at room temperature. The mixture was stirred at room temperature for 30 minutes and then placed into an ice bath, where EDC hydrochloride (445 mg, 2.32 mmol) and HOBt (313 mg, 2.32 mmol) were added and stirred at room temperature. After 18 hours, TLC (4:1 DCM:MeOH, $R_f$ = 0.35) showed consumption of starting material and formation of a pair of closely separated faster moving spots. The mixture was concentrated and briefly purified by column chromatography (silica gel; 12–20% MeOH in DCM), and the pure product 2-deoxy-2-*N*-pentynoy-lamide-D-mannopyranose (ManNAlk, **2**) was isolated at 16% MeOH in DCM as colorless syrup (523 mg, 87%). The purified product was characterized by ESI-TOF (m/z [M+Na]$^+$ calc. for C$_{11}$H$_{17}$NO$_6$Na 282.0954; found 282.0950) analysis, and we proceeded to the next step of the reaction.

To a solution of **2** (500 mg, 1.92 mmol) in anhydrous DMF (8 mL) and anhydrous NEt$_3$ (2 mL), 9-iodophenanthrene (645 mg, 2.12 mmol), Pd(PPh$_3$)$_2$Cl$_2$ (40 mg, 0.06 mmol), and CuI (37 mg, 0.2 mmol) were added and stirred overnight at 60 °C. TLC (4:1 DCM:MeOH, $R_f$ = 0.48 and 9:1 DCM:MeOH, $R_f$ = 0.25) showed complete consumption of the starting material and the formation of a faster-moving spot. The reaction mixture was evaporated to dryness and the crude mixture was purified by column chromatography. The pure product **3** was isolated at 10% MeOH in DCM as a brown foam (596 mg, 71%). ESI-TOF HRMS analysis confirmed the formation and isolation of the desired product **3** (m/z [M+Na]$^+$ calc. for C$_{25}$H$_{25}$NO$_6$Na 458.1580; found 458.1585). Propionic anhydride (1 mL, 7.6 mmol) was added to a stirring solution of **3** (550 mg, 1.26 mmol) in dry pyridine and dry DCM (1:1 *v/v*, 6 mL) and stirred overnight. The next day, TLC (9:1 DCM:MeOH, $R_f$ = 0.40 and 3:2 Hexane:EA, $R_f$ = 0.75) showed

complete consumption of the starting material and formation of a pair of closely separated faster-moving spots correlating with an anomeric mixture. The reaction mixture was evaporated under vacuum. The crude mixture was dissolved in DCM and subsequently washed with 1(N) HCl, saturated NaHCO₃, and brine solution. The combined washings were collected and dried over Na₂SO₄, and then evaporated under vacuum. The obtained crude product was purified by column chromatography (30% EA in Hexane) to obtain the upper spot (**4β**) as a white solid (178 mg, 21%). M. Pt. (Hexane/EA) = 113–115 °C; $^1$H NMR of upper spot (600 MHz, CDCl₃): δ 8.69–8.63 (m, 2H, Ar$H$), 8.40 (m, 1H, Ar$H$), 7.97 (s, 1H, Ar$H$), 7.85 (d, 1H, $J$ = 8.4 Hz, Ar$H$), 7.70–7.63 (m, 3H, Ar$H$), 7.58 (t, 1H, $J$ = 7.6 Hz, Ar$H$), 6.1 (d, 1H, $J$ = 1.7 Hz, H-1), 5.96 (1H, $J$ = 9.0 Hz, N$H$), 5.37 (dd, 1H, $J$ = 10.2 Hz, 4.6 Hz, $H$-3), 5.21 (t, 1H, $J$ = 10.3 Hz, $H$-4), 4.72 (m, 1H, $H$-2), 4.19 (m, 1H, $H$-6/6′), 4.06–4.00 (m, 2H, $H$-5, $H$-6/6′), 3.01–2.94 (m, 2H, C$H_2$), 3.72–2.69 (m, 2H, C$H_2$), 2.47–2.42 (m, 2H, C$H_2$), 2.34–2.25 (m, 4H, 2× C$H_2$), 2.22–2.10 (m, 2H, C$H_2$), 1.20–1.18 (t, 3H, $J$ = 7.6 Hz, C$H_3$), 1.11–1.07 (m, 6H, 2× C$H_3$), 0.97-0.94 (t, 3H, $J$ = 7.6 Hz, C$H_3$). $^{13}$C NMR (150 MHz, CDCl₃): δ 173.9 (C=O), 173.4 (C=O), 173.1 (C=O), 171.5 (C=O), 171.1 (C=O), 137.8, 131.3, 131.1, 130.1, 130.0, 128.5, 127.2, 127.0, 126.9, 126.8, 122.7, 122.5, 119.6, 92.5 (C-C-Ar), 91.6 (C-1), 79.9 (-C-C-Ar), 70.2 (C-5), 68.7 (C-3), 65.3 (C-4), 61.8 (C-6), 49.6 (C-2), 35.7 (CH₂CO), 27.4 (CH₂CH₃), 27.3 (2×C, CH₂CH₃), 27.2 (CH₂CH₃), 15.9 (CH₂C), 9.0 (CH₂CH₃), 8.9 (CH₂CH₃), 8.8 (CH₂CH₃), 8.7 (CH₂CH₃). HRMS (ESI-TOF): $m/z$ [M+Na]$^+$ calcd. for C₃₇H₄₁NO₁₀Na 682.2628; found 682.2629.

Continued elution resulted in a mixture of isomers (245 mg, 30%) followed by the pure major isomer (**4α**) as a colorless syrup (352 mg, 42%, combined yield 775 mg, 93%). $^1$H NMR of lower spot (600 MHz, CDCl₃): δ 8.69–8.62 (m, 2H, Ar$H$), 8.42 (m, 1H, Ar$H$), 7.97 (s, 1H, Ar$H$), 7.83 (dd, 1H, $J$ = 7.8 Hz, 1 Hz, Ar$H$), 7.71-7.62 (m, 3H, Ar$H$), 7.58 (ddd, 1H, $J$ = 7.9 Hz, 6.9 Hz, 1.1 Hz, Ar$H$), 6.09 (d, 1H, $J$ = 8.8 Hz, N$H$), 5.89 (d, 1H, $J$ = 1.9 Hz, H-1), 5.19 (t, 1H, $J$ = 9.8 Hz, $H$-4), 5.11 (dd, 1H, $J$ = 9.9 Hz, 4.1 Hz, $H$-3), 4.85 (ddd, 1H, $J$ = 5.0 Hz, 4.0 Hz, 1.7 Hz, $H$-2), 4.25 (dd, 1H, $J$ = 12.4 Hz, 5.3 Hz, $H$-6/6′), 4.10 (dd, 1H, $J$ = 12.4 Hz, 2.4 Hz, $H$-6/6′), 3.82 (m, 1H, $H$-5), 3.01–2.95 (m, 2H, C$H_2$), 2.77–2.70 (m, 2H, C$H_2$), 2.53–2.08 (m, 8H), 1.12–1.08 (m, 6H, 2× C$H_3$), 1.03–0.99 (t, 3H, $J$ = 7.6 Hz, CH₃), 0.91–0.87 (t, 3H, $J$ = 7.6 Hz, CH₃). $^{13}$C NMR (150 MHz, CDCl₃): δ 173.9 (C=O), 173.4 (C=O), 173.1 (C=O), 171.9 (C=O), 171.6 (C=O), 131.7, 131.3, 131.2, 130.0, 129.99, 128.4, 127.3, 127.0, 126.97, 126.9, 126.8, 122.7, 122.5, 119.7, 92.7 (C-C-Ar), 90.6 (C-1), 79.7 (-C-C-Ar), 73.5 (C-5), 71.2 (C-3), 65.2 (C-4), 61.7 (C-6), 49.7 (C-2), 35.7 (CH₂CO), 27.4 (CH₂CH₃), 27.3 (CH₂CH₃), 27.2 (CH₂CH₃), 27.0 (CH₂CH₃), 16.0 (CH₂C), 9.0 (CH₂CH₃), 8.9 (CH₂CH₃), 8.7 (CH₂CH₃), 8.3 (CH₂CH₃). HRMS (ESI-TOF): $m/z$ [M + NH₄]$^+$ calcd. for C₃₇H₄₅N₂O₁₀ 677.3074; found 677.3076.

### Fluorescence Spectroscopy

Fluorescence excitation and emission spectra were acquired on a Horiba Jobin Yvon Fluorolog FL-3 fluorimeter using a 3 × 3 × 10 mm quartz cuvette. Excitation spectra were recorded from 300 to 430 nm (1 nm step, 0.5 mm slit width, 0.5 s integration time) at the emission maximum of 437 nm (0.5 mm slit width). Emission spectra were recorded from 385 to 600 nm (1 nm step, 0.5 mm slit width, 0.5 s integration time) using an excitation wavelength of 380 nm (0.5 mm slit width). Emission intensity was corrected for lamp power and detector sensitivity.

### Cell culture

HeLa and mouse embryonic fibroblast (MEF) cells were cultured in DMEM (Gibco, 10567022) supplemented with 10% (for HeLa) or 15% (for MEFs) fetal bovine serum (FBS), penicillin (100 U/mL), and streptomycin (1 mg/mL). HeLa cells were kept at 37 °C in a humidified incubator under 5% carbon dioxide (CO₂) atmosphere, and MEFs were kept at 37 °C in an incubator with 3% oxygen (O₂) atmosphere. LA-N-2 and SH-SY5Y cells were cultured in DMEM/F-12 medium with L-glutamine (Gibco, 11320033) supplemented with 10% FBS, penicillin (100 U/mL), and streptomycin (1 mg/mL), and were kept at 37 °C in a humidified incubator under 5% CO₂ atmosphere. NIH 3T3 cells were cultured in DMEM (Gibco,

10567022) supplemented with 10% donor bovine serum (DBS), penicillin (100 U/mL), and streptomycin (1 mg/mL), and were kept at 37 °C in a humidified incubator under 5% CO₂ atmosphere. Non-malignant MCF10A (tissue: mammary gland/breast, cell type: epithelial, CRL-10317) and malignant MCF7 (tissue: mammary gland/breast; derived from a metastatic site: pleural effusion, cell type: epithelial, HTB-22) breast cancer cells were purchased from ATCC (American Type Culture Collection, LGC Standards) and propagated in monolayer culture using standard mammalian cell culture techniques as described in the ATCC protocols. MCF10A cells were incubated at 37 °C under an atmosphere of air, 95% humidity, and 5% CO₂ in a nutrient-rich MEBM medium (Lonza, CC-3151) along with the additives obtained from the EGM BulletKit kit (Lonza/Clonetics catalog No. CC-3150) containing h-EGF-β (human epidermal growth factor), BPE (bovine pituitary extract), hydrocortisone and human recombinant insulin. Per the ATCC protocol the gentamycin–amphotericin B mix provided with the kit was not used. To prepare the complete growth medium, 100 ng/ml of cholera toxin (Sigma Aldrich, C8052) was added. MCF7, HEK 293T and HepG2 cells were cultured in EMEM medium (ATCC, 30-2003) supplemented with 2 mM L-glutamine, 0.01 mg/ml human recombinant insulin (only for MCF7 cells, Sartorious, BE02-033E20), 10% FBS, 1% penicillin (100 U/mL), and streptomycin (1 mg/mL), and were kept at 37 °C under an atmosphere of air, 95% humidity, and 5% CO₂. LecCHO (tissue: Ovary; derived from Chinese hamster; Cell type: epithelial, CRL-1735) cells were purchased from ATCC (LGC Standards) and propagated in monolayer culture using standard mammalian cell culture techniques as described in the ATCC protocols. LecCHO cells were cultured in the Alpha minimum essential medium with ribonucleosides and deoxyribonucleosides (MEMα, Gibco, 12571063) supplemented with 10% FBS, penicillin (100 U/mL), and streptomycin (1 mg/mL), and were kept at 37 °C under an atmosphere composed of air, 95% humidity, and 5% CO₂. AML12 (tissue: Liver; derived from Mouse; Cell type: Hepatocyte, CRL-2254) cells were purchased from ATCC (LGC Standards) and propagated in monolayer culture using standard mammalian cell culture techniques as described in the ATCC protocols. AML12 cells were cultured in the DMEM:F-12 medium with L-glutamine (Gibco, 11320033) supplemented with 1% Insulin-transferrin-Selenium (ITS-G, Gibco, 41400045), 40 ng/ml Dexamethasone (10 mM in DMSO, Gibco: A13449), 10% FBS, 1% penicillin (100 U/mL), and streptomycin (1 mg/mL), and were kept at 37 °C under an atmosphere composed of air, 95% humidity, and 5% CO₂. Lymphocytes collected from the spleens of wild-type mice were cultured in complete RPMI media (Thermo Fisher, 11875093) supplemented with 10% FBS, 0.1% BME, 1% glutamine, 1% penicillin (100 U/mL), and streptomycin (1 mg/mL), and were kept at 37 °C in a humidified incubator with 5% CO₂.

For imaging experiments, cells were seeded at a density of 10k cells per well in a 4-well chambered coverglass slide, 100k cells/well in a 6-well cell culture plate, 5k cells/well in a 96-well cell culture plate, and 500k cells in a 10 cm plate. Cells were allowed to settle overnight and then an appropriate amount of sugar was added for the desired final concentration from a 100 mM stock. Alternatively, the equivalent volume of DMSO or an equal concentration of 9-iodophenanthrene was added as a negative control.

### Cell Lysate collection

After the appropriate time, cells were harvested by physical scraping, counted, and centrifuged at 10,000 × $g$ for 5 min in 1.5 mL sterile RNAse and DNAse-free Eppendorf tubes. The supernatants were removed carefully, and the collected cell pellets were stored at 80 °C for future use. Cell pellets were lysed with chilled RIPA lysis buffer in an ice bath for 10 min with occasional vortexing, and then centrifuged at 4 °C at 10,000 × $g$ for 10 min in 1.5 mL sterile RNAse and DNAse-free Eppendorf tubes. Cell lysates were stored at −20 °C for temporary storage or −80 °C for longer storage until use.

### Mice

The animals were maintained according to the animal protocol #K023-LCBB-22 approved by the NIDDK Animal Care and Use Committee,

National Institutes of Health, USA and the C57Bl/6J-Mgea5$^{tm2Jah}$/cre mice (3-Male, 3-Female) were analyzed between 10-12 weeks of age.

## Confocal microscopy

Cells were grown to sub-confluency and the treatment compound or negative control vehicle was added and incubated for the required amount of time. After the appropriate time, SYTO Deep Red Nuclear Dye (Thermo Fisher Scientific, S34900) or LysoTracker Deep Red (Thermo Fisher Scientific, L12492) was added, and the cells were incubated for an additional 1 hour. Cells were washed thrice with PBS and fixed in 2% PFA in 1XPBS for 15 minutes at 37 °C. For LAMP1 colocalization studies, cells were washed thrice with 1XPBS, blocked with 0.1% BSA in 1XPBSTw for 1 hour, then incubated with the LAMP1 primary antibody (1:100 dilution in 0.1% BSA) for 1 hour at room temperature and washed thrice with 1XPBSTw. They were then incubated with Alexa Fluor-conjugated secondary antibody (1:500 dilution in 0.1% BSA in 1XPBS) for 1 hour at room temperature. Cells were again washed thrice with 1 × PBSTw prior to imaging. When required, samples were then mounted with AlexaFluor 594 WGA (Thermo Fisher Scientific, W11260) or ConA Texas red (Thermo Fisher Scientific, C825) or subjected to Triton X-100, saponin, or cold MeOH permeabilization for 30 minutes at room temperature, followed by washing with 1XPBSTw (3 washes, 5 minutes each). Confocal fluorescence images were acquired on an LSM 700 laser scanning confocal microscope (Zeiss) equipped with a 40× water immersion objective. The lysotacker experiment images in Fig. 3A were captured using a Leica STELLARIS confocal microscope equipped with a 100× oil immersion objective. MM-JH-2 was visualized using 405 nm laser excitation, AlexaFluor 488 was visualized using 488 nm laser excitation (for the click reaction product with AlexaFluor 488 azide), and Deep Red conjugated stains were visualized using 560 nm laser excitation. Microscope settings were identical for all images within each figure. Fluorescence image analysis was performed in the FIJI image processing software. The red signal corresponds to a DNA intercalated dye, which serves as a normalization control. The blue signal corresponds to the specific incorporation of MM-JH-2. This ratio is expected to increase with increased labeling, and the absolute value of the ratio is somewhat arbitrary. Signal intensities exceeding a constant threshold were quantified for each channel using Fiji software. Figures show representative images.

## In-gel fluorescence scanning

Following SDS-PAGE gel separation, the gel was washed with water 10 minutes for 3 times and scanned on a Molecular Imager FX (Bio-Rad) using a 405-nm laser for excitation. For Coomassie staining, the gel was incubated with Coomassie blue solution for 1 hours followed by destaining using the destaining solution (40% methanol, 10% glacial acetic acid) for overnight, followed by water for an additional 10 minutes for 3 times prior to scanning. The gel was scanned on a Molecular Imager FX (Bio-Rad) using a 405-nm laser for excitation.

## siRNA treatment experiment

HeLa cells were seeded in a 4-well chambered coverglass slide (10k cells/well) or a 6-well cell culture plate (100k cells/well), and cultured in antibiotic-free complete growth media. For siRNA experiments: DMEM (Gibco, A14430 without phenol red) for HeLa, and DMEM/F-12 (Gibco, 11320033) for SH-SY5Y, respectively, were supplemented as above without antibiotics. The following siRNA reagents were purchased from Horizon, a PerkinElmer company: transfection reagent (T-2001, Dharmacon), *ST3GAL5* siRNA construct (human) (L-011546-00-0005, Dharmacon), and control siRNA (4390843, Dharmacon, Lafayette, CO). Lyophilized siRNA duplex was resuspended in 250 μL of RNAse-free water to make a 20 μM stock solution. HeLa cells were seeded either in 6-well plates or in 4-well chambered coverglass slides in normal growth medium without antibiotics and grown to ~50% confluency in the incubator at 37 °C, 5% CO$_2$. For transfections in the 4-well chambered coverglass slide, either 0.4 μL siRNA duplex (10 nM siRNA),1 μL siRNA duplex (25 nM siRNA), or 2 μL siRNA duplex (50 nM siRNA) was diluted into 80 μL of antibiotic-free,

unsupplemented growth media and incubated at room temperature for 5 minutes. Separately, 5 μL siRNA transfection reagent was diluted in 75 μL of antibiotic-free, unsupplemented media and incubated at room temperature for 5 minutes. The 80 μL of siRNA duplex was mixed with the 80 μL of transfection reagent and incubated at room temperature for 30 minutes. After 30 minutes, this mixture was added to 640 μL of antibiotic-free full growth media and added to the plated cells, followed by incubation for 24 hours at 37 °C in a 5% CO$_2$ incubator. For transfections in the 6-well plate, either 1 μL of siRNA duplex (10 nM siRNA), 2.5 μL of siRNA duplex (25 nM siRNA), or 5 μL of siRNA duplex (50 nM siRNA) was diluted into 200 μL of antibiotic-free unsupplemented growth media and incubated at room temperature for 5 minutes. Similarly, 15 μL of siRNA transfection reagent was diluted into 185 μL of antibiotic-free and unsupplemented media and incubated at room temperature for 5 minutes. The 200 μL of siRNA duplex was mixed with the 200 μL of transfection reagent and incubated at room temperature for 30 minutes. After 30 minutes, this mixture was added to 1600 μL of antibiotic-free full growth media and added to the plated cells, followed by incubation for 24 hours at 37 °C in a 5% CO$_2$ incubator. After 24 hours, the procedure was repeated with the addition of DMSO or 25 μM MM-JH-2 to the antibiotic-free growth medium and cells were incubated for an additional 72 hours at 37 °C in a 5% CO$_2$ incubator. After 72 hours of incubation, cells were either fixed with PFA for confocal imaging or isolated by trypsinization and processed as described for immunoblotting.

## Immunoblotting

Pelleted cells were lysed in RIPA lysis buffer in an ice bath for 10 minutes with periodic shaking and then centrifuged at 4 °C at 10,000 × g for 10 minutes in 1.5 mL Eppendorf tubes. Cell lysates were stored at −20 °C degrees for temporary storage or −80 °C for longer storage until use. Protein concentrations were determined by BCA assay (Pierce, Thermo Fisher Scientific) and normalized to lowest protein concentration using RIPA buffer. 25 μg of protein was mixed with LDS sample loading buffer with β-mercaptoethanol (BME). Proteins were analyzed by SDS PAGE on 4–12% Bis-tris gels (Invitrogen) in 1 X MOPS running buffer after which they were transferred to a 0.2 nm nitrocellulose membrane using an iBlot 2 Gel Transfer Device (Invitrogen IB21001). Membranes were blocked with Odyssey PBS blocking buffer (Li-Cor) at room temperature for 1 hour prior to incubation with the appropriate primary antibody (1:1000 dilution) in Odyssey PBS blocking buffer with 0.1% tween-20 (PBSBBTw, Li-Cor) at 4 °C overnight. The next day blots were washed three times in PBSTw for 10 minutes each and incubated with appropriate secondary Odyssey antibodies (1:10,000 dilution, Li-Cor) in Odyssey PBSBBTw (Li-Cor) at room temperature for 1 hour before being washed thrice with PBST and imaged on an Odyssey instrument (Li-Cor). Blots were quantified using the Image Studio software package (Li-Cor).

## MALDI Mass spectrometry

Lipids were extracted from aliquots of 2 × 10$^5$ cells. A mix of chloroform/methanol/H$_2$O (1:2:0.75; $v/v/v$) was added, sonicated for 10 minutes, and centrifuged (14,000 × g, 5 minutes). The supernatant was collected, and these two steps were repeated once more. Mixtures of chloroform/methanol (1:1; $v/v$) and chloroform/methanol (2:1; $v/v$) were sequentially added and sonicated for 10 minutes. Samples were centrifuged and the supernatants were collected and pooled after each step. Pooled supernatants were dried using a SpeedVac (ThermoScientific). The dried total lipid extract was resuspended in 800 μL of butanol and 800 μL of water, sonicated in a water bath for 10 minutes, and centrifuged (14,000 × g, 5 minutes). The blue, fluorescent upper organic layer was collected using a glass Pasteur pipette and the extracted gangliosides were again dried using a SpeedVac (ThermoScientific). Collected samples were dissolved in 50% ACN, 2.5% TFA to a concentration of 1 mg/ml, and 2 μl were dried on the MALDI target. Samples were overlaid with 2 μl of 20 mg/ml DHB matrix in 50% ACN, 2.5% TFA and analyzed in positive reflectron mode using an AutoFlex III (Bruker). Calibration was performed using Bruker Peptide standards (std

https://doi.org/10.1038/s42004-025-01685-x **Article**

dev 8ppm). Peaks of interest were subjected to LIFT analysis for fragmentation.

## Bafilomycin A1 treatment study

HeLa cells were seeded in a 4-well chambered covered glass slide (10k cells/well) and cultured as described earlier. Different concentrations of Bafilomycin A1 (Cell Signaling, 54645) were added 12 hours after initial cell seeding and cells were incubated for another 6 hours under standard growth conditions. Cells were treated with 25 µM MM-JH-2 or an equal volume of DMSO as a negative control and incubated for an additional 72 hours. Cells were treated with either SYTO Deep Red Nucleic Acid stain or LysoTracker Deep Red and incubated for an additional 1 hour, followed by washing and fixation for confocal fluorescence imaging.

## Chloroquine treatment study

HeLa cells were seeded in a 4-well chambered covered glass slide (10k cells/well) and cultured as described earlier. Chloroquine (Sigma-Aldrich, C6628), or DMSO as a negative control, was added 24 hours after initial cell seeding and cells were incubated in a 37 °C $CO_2$ incubator for another 12 hours. Cells were treated with 25 µM MM-JH-2 and incubated in a 37 °C $CO_2$ incubator for an additional 4 hours. Cells were treated with Lyso-Tracker Deep Red and incubated in a 37 °C $CO_2$ incubator for an additional 1 hour, followed by washing and fixation for confocal fluorescence imaging.

## eGFP-transfection experiment

HeLa cells were plated in a 4-well chambered coverglass slide (10k cells/well) and cultured as described earlier. For eGFP transfection, DMEM (Gibco, 10567022) was supplemented with 10% fetal bovine serum (FBS) without antibiotics. Plasmids for expression of eGFP-caveolin (WT) as well as WT and K44A dynamin were a generous gift from Dr. Justin Taraska[62]. Cells were transiently transfected with plasmid DNA 24 hours after initial seeding using Lipofectamine 2000 (Thermo Fisher Scientific, 11668019) according to the manufacturer's protocol. Briefly, 0.8 µL plasmid DNA (1 µg/µL) and 2.4 µL of transfection reagent were separately diluted into 50 µL of Opti-MEM (Thermo Fisher Scientific, 31985070). After 5 min incubation, the diluted DNA and transfection reagent were thoroughly mixed and incubated for 20 min at room temperature before being added to the cells. A total of 500 µL of antibiotic-free media containing 100 µL of transfection mixture was added to each well, and cells were incubated in a 37 °C $CO_2$ incubator for 6 hours. Cells were treated with 25 µM MM-JH-2 and incubated in a 37 °C $CO_2$ incubator for an additional 4 hours followed by washing and fixation for confocal fluorescence imaging.

## CuAAC (click) reaction

CuAAC reactions were performed using a Click-iT Cell Reaction kit (Thermo Fisher Scientific, C10269). To fixed cells, a cocktail of 440 µL Click-iT reaction buffer (from a stock of 1:10 diluted solution of component A), 10 µL of $CuSO_4$ solution, 50 µL of Click-iT cell buffer additive (from a stock of component C diluted in 4 mL of deionized water) and 2.5 µL of Alexa Fluor 488 azide (from a 1 mM stock solution in DMSO) was added. Samples were incubated in the dark for 30 min at room temperature and washed three times with PBS.

## Raman spectral imaging

HeLa cells were seeded at a density of 10k cells/well in Nunc LabTek 4-well chambered #1.5 coverglass slides (ThermoFisher Scientific) and cultured as described earlier. Raman spectral imaging was conducted on a home-built Raman microscope[20]. Excitation was provided by the 514-nm line of an air-cooled argon-ion laser (Melles-Griot, 35-MAP-431-200) at 124 mW for cell mapping or the 633-nm line of a JDSU HeNe laser (JDSU 1135P) at 10 mW for pure MM-JH-2. The beam was directed through a beam expander (ThorLabs) into the rear port of an Olympus IX-71 inverted microscope, through a laser line cleanup filter (514 line only, LL01-514-25, Semrock), and directed by a RazorEdge dichroic beamsplitter (LPD01-514RU or LPD02-633RU, Semrock) into a 60×/1.42 NA oil immersion objective

(PLAPON60XO, Olympus). Backscattered light was collected through the objective and passed through an emission filter (NF03-514E-25 or LP02-633RU-25, Semrock) before being focused into an iHR320 spectrometer (Horiba) configured with a 50 µm entrance slit and a 600 line/mm grating. Spectra were recorded on a Symphony II back-illuminated deep-depleted liquid $N_2$ cooled CCD (Horiba) binned from pixels 124 to 132 in the $y$-dimension to achieve confocality. The sample was scanned using a Scan IM 120 × 80 motorized stage (Märzhauser-Wetzlar) at a step size of 400 nm. Individual spectra were recorded with a 500-ms accumulation time at each spot in cell mapping experiments, and 256 1-s accumulations were averaged for the isolated compound. Widefield epifluorescence images were acquired using an OBIS LX 405 nm laser (Coherent) expanded through a 10×/0.25 NA air objective (PLN10X, Olympus) and directed into the rear port of the microscope. Switching between Raman and widefield excitation lasers was accomplished *via* a flip mirror. The laser was focused into the back aperture of the same 60×/1.42 NA oil immersion objective through a 405-nm long-pass dichroic (Di02-R405, Semrock) and fluorescence was collected through a 420-nm long-pass emission filter (BA420, Nikon). Images were recorded on an Infinity 3S-1URM CCD (Lumenera). Bright-field images were acquired using the same objective and CCD.

## Raman spectral analysis

All Raman shifts were calibrated daily using a spectrum of neat cyclohexane fit to a linear combination of Gaussian-Lorentzian functions in the LabSpec6 software package (Horiba). The resulting peak positions were fit to literature values to generate a linear correction for the Raman shift values. Raman spectra of pure MM-JH-2 in DMSO were background-subtracted using a spectrum of DMSO collected with identical parameters in an adjacent well of the 18-well chambered coverslip (C18SB-1.5H, Cellvis) at the same $z$-position. A four-degree polynomial baseline correction was performed on the resulting spectrum using the baseline correction algorithm in LabSpec6.

## Raman Map Image analysis

Raman maps were generated from spectral data by integrating over spectral regions of interest in LabSpec6 (763–794 $cm^{-1}$ for nucleotides, 2200–2245 $cm^{-1}$ for alkyne, and 2835–2856 $cm^{-1}$ for lipids). A linear baseline subtraction was applied over each integrated region. Subsequent image analysis was performed using ImageJ. For maps that were collected only over the area of the cell (i.e., non-rectangular), the empty areas of each map were filled with noise to match the standard deviation of the noise in the mapped area using the Specified Noise function. Background subtraction was performed for each map using a Gaussian-blurred image with a radius of 100 pixels. A mask was generated by subtracting a 100 pixel radius Gaussian blurred image from a 1 pixel radius Gaussian blurred image, applying an automatic threshold using the Li method, and converting the resulting image to a mask. Finally, the mask was applied to the background-subtracted image.

## Widefield Epifluorescence Image Analysis

Widefield epifluorescence images of cells selected for Raman spectral imaging were background-subtracted using a Gaussian-blurred image with a radius of 25 pixels. A mask was generated by applying an automatic threshold to the background-subtracted image using the Li method and converting the resulting image to a mask. The final image was generated by applying the mask to the background-subtracted image.

## 3-(4,5-Dimethylthiazol-2-yl)-2,5-diphenyltetrazolium (MTT) Assay

Cells were seeded in one half of each 96-well culture plate (5k cells/well) and cultured as described earlier, while the other half of the wells were left blank, containing only media with no cells. After 24 hours, the media was replaced with 200 µL of fresh media supplemented with MM-JH-2 (0, 5, 10, 25, 50, 100, 250 and 500 µM) and cells were incubated for an additional 72 hours.

Thereafter, 20 µL of MTT labeling reagent (Sigma Aldrich, MTT Kit, 11465007001) was added to each of the wells and the plate was incubated in a 37 °C $CO_2$ incubator in the dark for 4 hours. The formazan crystals formed by the cells were dissolved using 100 µL of solubilization buffer (Sigma Aldrich, MTT Kit, 11465007001) in a 37 °C $CO_2$ incubator overnight. Absorbance readings were recorded at 570 nm using 630 nm as a reference wavelength on a POLARstar Omega microplate reader (BMG). Reduced formazan was quantified using a formazan standard and background correction was performed using blank wells.

The results represent the average of 6 replicates from 5 independent experiments performed over multiple days. The percentage of cell viability was calculated using the equation:

$$\text{Cell viability} (\%) = (\text{OD of treatment}/\text{OD of control}) \times 100$$

### Flow cytometry

Spleens were harvested from mice, tissue was disrupted mechanically in complete RPMI media, passed through a 70 µm cell strainer, and washed with complete media. Cells were counted after ACK lysis using a Nexcelon Cellometer. $1.5 \times 10^3$ cells were seeded in a 96-well plate and incubated with different concentrations of MM-JH-2 (25, 50 and 100 µM) or DMSO for 2 or 4 hours. Cells were washed thrice with flow cytometry staining buffer (Thermo Fisher scientific, 00-4222-26) and blocked with 100 µL of flow cytometry blocking buffer (anti Mouse CD16/CD32 blocking Thermo Fisher Scientific, 14-0161-85, 1:100 dilution in flow cytometry staining buffer) for 15 minutes at 4 °C. For five-color flow cytometry analysis, cells were incubated with APC-CD4+ (Thermo Fisher Scientific, 17-0042-82), FTIC-CD8+ (Thermo Fisher Scientific, 11-0081-85), and PE-B220+ (Thermo Fisher Scientific, 12-0452-82) for 30 minutes at room temperature (antibodies were used in a 1:200 dilution in flow cytometry staining buffer). To minimize nonspecific binding, cells were washed three times in flow cytometry staining buffer and finally resuspended in 150 µL of flow cytometry staining buffer, followed by the addition of 5 µL PI solution (Thermo Fisher Scientific, 00-6990-42) for live/dead differentiation. Samples were analyzed on a LSRFortessa flow cytometer equipped with 355 nm, 407 nm, 488 nm, 532 nm, and 640 nm laser lines and a high-throughput sampler system (BD Biosciences). Data were recorded using BD FACS Diva (version 8.0) and analyzed in the Flowjo v10.9.0 software package (both from BD Biosciences). The results represent the average of 6 replicates from 6 independent experiments from two different mice.

Debris-free single live cells (identified by the absence of propidium iodide staining) were classified using cell physical parameters (FSC/SSC). Live single cells were further gated for B cells as B220+, CD4+ T cells as B220-CD8-CD4+ cells, CD8+ T cells as B220-CD4-CD8+ cells, whereas remaining cells containing heterogeneous cell populations as B220-CD4-CD8- live single cells. MM-JH-2 was excited by the UV laser line and detected using a 450/50 nm band pass filter. fluorescence minus one (FMO) control containing cell surface antibodies but not MM-JH-2 was used to define the presence of dye and as correlate for uptake of MM-JH-2. Data is presented as % positive subsets for the probe as well as mean fluorescence intensity.

### Statistics

Prism (version 10.3.1 for MacOS, GraphPad) was used for all statistical analyses. A two-way ANOVA test was used to determine significance as indicated in the figure legends. P-values less than 0.05 were considered statistically significant. For confocal fluorescence images, co-localization analysis was performed in Fiji (ImageJ version 2.14.0/1.54 f) using the co-localization plugin. Pearson's correlation coefficients between MM-JH-2 and WGA, ConA, LAMP1, Lysotracker, or eGFP were measured using two regions of interest (ROI) per image.

### Data availability

The data that support the findings of this study are available within the paper and the Supplementary Information.

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

## Acknowledgements

MMM, DB, LA and JAH thank members of the Hanover lab for discussion, advice, and valuable suggestions. This work was conducted in the Laboratory of Cell and Molecular Biology (LCMB) section in NIDDK, National Institutes of Health, and supported by the NIDDK, National Institutes of Health. This research was supported by the Intramural Research Program of the National Institute of Diabetes and Digestive and Kidney Diseases (NIDDK) within the National Institutes of

Health (NIH). The contributions of the NIH author(s) were made as part of their official duties as NIH federal employees, are in compliance with agency policy requirements, and are considered Works of the United States Government. However, the findings and conclusions presented in this paper are those of the author(s) and do not necessarily reflect the views of the NIH or the U.S. Department of Health and Human Services. MMM, DB, LA and JAH thanks intramural research program grant at the NIDDK, NIH (**DK060101-18**). MDW and JCL are supported by the Intramural Research Program at the NIH, NHLBI (ZIA HL006236-06). The views, information or content, and conclusions presented do not necessarily represent the official position or policy of, nor should any official endorsement be inferred on the part of, the NIDDK, or NHLBI, or Clinical Center, the National Institutes of Health, or the Department of Health and Human Services. We thank Dr. Justin Taraska (NHLBI, NIH) for his generous gift of eGFP-Dynamin1 (WT), eGFP-dynamin1 K44A mutant, and eGFP-caveolin (WT) plasmid DNA. We also thank the NIDDK ALMIAC imaging facility for confocal microscopy. We are thankful to the NHLBI Flow Cytometry Core facility, especially Dr. Pradeep K. Dagur for the use of the Flow Cytometry facility. The graphical abstract was created with BioRender.com. DK060101-18 J. A. H. ZIA HL006236-06 J. C. L.

## Author contributions

J.A.H. conceived of the project, supervised the study, wrote and edited the manuscript, and arranged necessary funding. M.M.M. conceptualized, synthesized and characterized the chemical compounds, performed most of the experiments, analyzed data, wrote the manuscript and coordinated with other co-authors. D.B. performed confocal imaging, assisted in the synthesis of the chemical compound, analyzed data, wrote and edited the manuscript. L.K.A. harvested mouse spleen, oversaw the flowcytometry experiment, provided invaluable feedback on the project, wrote and edited the manuscript. M.D.W. and J.C.L. conducted Raman spectroscopy and wrote a section of the manuscript. S.K.D. performed MALDI mass spectrometric analysis of the ganglioside extracts and provided invaluable feedback on the MALDI mass analysis.

## Funding

## Competing interests

The authors declare no competing interests. The authors are actively looking for partners to commercialize MM-JH-2.
