## [Transparent Peer Review file · Communications Chemistry]

A Dual Fluorescent-Raman Bioorthogonal Probe for Specific Biosynthetic Labeling of Intracellular Gangliosides

Corresponding Author: Dr John Hanover

Version 0:

Reviewer comments:

Reviewer #1

(Remarks to the Author)

In this manuscript, the authors present a new fluorescent/Raman labeling method for cellular glycoside lipids. They extended the alkyne-tagged mannoside derivative ManNalk with phenanthrene, resulting in a new mannoside analog with both Raman and fluorescent properties, named phenanthrene-9-Pr4ManNAIk (MM-JH-2). This probe was successfully incorporated into cells and metabolized to a ganglioside GM3, as confirmed by MALDI-MS analysis. This work is thorough and includes various detailed analyses. However, the concept of this study is somewhat difficult to understand. Using an SRS microscope, the direct visualization of ManNalk with a tiny alkyne tag was already achieved in 2014 (Hong, S. et al *Angew. Chemie Int. Ed.* 2014, 53, 5827). Therefore, I could not understand the potential of MM-JH-2 with a bulky fluorescent/Raman tag. Moreover, if MM-JH-2 can be visualized using the fluorescent imaging, Raman imaging might not be necessary. What is the advantage of dual Raman/fluorescent dyes? To visualize small molecules, tiny Raman dyes are useful to avoid the decrease in bioactivity caused by introducing large fluorescent dyes. I'm afraid the introduction of large alkyne-phenanthrene tag might affect the native metabolism of mannoside, leading to the accumulation of the probe in acidic compartments. Additionally, the authors employed spontaneous Raman imaging, which is usually disrupted by fluorescent signals. Why did the fluorescence not interfere with the Raman imaging? Did the authors consider the possibility that the Raman signal of alkyne-phenanthrene might be enhanced by resonance Raman scattering? If the new dye proves useful for resonance Raman imaging, this work could be significant for publication. Please check the RIE, the standard of Raman intensity, in different excitation wavelengths. However, in its current form, the manuscript may not be suitable for publication.

Reviewer #2

(Remarks to the Author)

This paper describes a fluorescence- and Raman-sensitive probe that specifically labels intracellular gangliosides. Gangliosides are glycolipids abundant in the nervous system and play an important role in neuronal maintenance and regeneration. Abnormal ganglioside metabolism contributes to pathological conditions such as neurodegenerative diseases and cancer, and specific visualization of gangliosides is very useful for elucidating the mechanisms of the above biological phenomena. In this study, the authors developed a mannosamine derivative (MM-JH-2) with fluorescence and Raman activity, which permeates the cell membrane and acts as a precursor for sialic acid biosynthesis in the cytoplasm and is preferentially incorporated into ganglioside biosynthesis, thereby specifically labeling gangliosides. MM-JH-2 is a novel, highly selective, and highly efficient ganglioside labeling molecule. Furthermore, MM-JH-2 has been shown to preferentially label malignant breast cancer cell gangliosides in breast cancer cells, to be highly efficient in labeling B cells and T cells, and to specifically label cancer cells at low concentrations. These results, which have been carefully and multidimensionally validated, strongly demonstrate the utility of MM-JH-2 in biological studies: gangliosides labeled with MM-JH-2 are not only fluorescent but also Raman-sensitive, and the specific vibrational bands of alkynes MM-JH-2 can be distinguished from other intracellular biomolecules by the specific vibrational band of alkyne, and can be applied as an unprecedented and useful probe.

The results presented in this paper are worthy of publication in *Communication Chemistry*. However, to improve the quality of the paper, the following points should be considered and modified.

1. a discussion of why labeling with MM-JH-2 proceeds in a ganglioside-specific manner would be helpful.
2. information on the analysis of molecular species of gangliosides labeled by MM-JH-2 (e.g., malignant breast cancer cells) could be added.

Reviewer #3

(Remarks to the Author)

This manuscript describes a dual-fluorescence Raman probe for specific biosynthetic labeling of gangliosides present in the membranes of nerve cells. This is an important topic, and significant results have been obtained. Although there are still many unknowns regarding the dynamics and functions of glycolipids within cells, the probe developed in this study is expected to lead to significant progress. However, there is a lack of sufficiently convincing explanations, and certain aspects are difficult to understand, particularly for researchers in the field of cell biology who are interested in this study.

1. Figure 1C: You need to explain the quantitative method. I do not understand why the ratio of MM-JH-2 staining (blue) to nuclear staining (red) exceeds 1.
2. Figure 2D: Does this mean that MM-JH-2 binds to the terminal sialic acid of ganglioside? Please provide a more detailed explanation, including the relationship between MM-JH-2 binding and ganglioside specificity.
3. Figure 3: Gangliosides are abundant in cell membranes. MM-JH-2 did not colocalize with WGA-Texas red, a cell surface marker. Why?
4. Is MM-JH-2 transiently labeled during the synthesis of gangliosides? Does the stability of MM-JH-2 binding to gangliosides differ depending on pH?
5. Considering that the labeling status of gangliosides varies depending on MM-JH-2 concentration, cell type, intracellular pH, etc., how about using a paper title that reflects this?
6. Figure 6: It would be even better if there were images of the cells.
7. There are several instances of odd wording and grammatical errors.

Version 1:

Reviewer comments:

Reviewer #1

(Remarks to the Author)

In the revised manuscript, I understood the unique properties of MM-JH-2, especially for the specificity for intracellular gangliosides. The Raman detection is an additional feature and not the main function of this probe. Considering the modified parts of this manuscript, this manuscript is now suitable for publication.

Reviewer #2

(Remarks to the Author)

The authors has succeeded in improving the quality of the manuscript by addressing the comments made by the reviewer. The reviewer recommend the publication of the manuscript in Commun Chem.

Reviewer #3

(Remarks to the Author)

The manuscript has been much improved.

Response to the reviewer's comments:

Reviewer #1:

In this manuscript, the authors present a new fluorescent/Raman labeling method for cellular glycoside lipids. They extended the alkyne-tagged manoside derivative ManNAIk with phenanthrene, resulting in a new manoside analog with both Raman and fluorescent properties, named phenanthrene-9-Pr4ManNAIk (MM-JH-2). This probe was successfully incorporated into cells and metabolized to a ganglioside GM3, as confirmed by MALDI-MS analysis. This work is thorough and includes various detailed analyses.

However, the concept of this study is somewhat difficult to understand. Using an SRS microscope, the direct visualization of ManNAIk with a tiny alkyne tag was already achieved in 2014 (Hong, S. et al *Angew. Chemie Int. Ed.* 2014, 53, 5827). Therefore, I could not understand the potential of MM-JH-2 with a bulky fluorescent/Raman tag.

We sincerely thank the reviewer for pointing out this important issue. The paper cited by the reviewer describes SRS imaging on K20 cells treated with Ac4ManNAIk. Their images in figure 4 showed a nuclear localization of the probe which they reasoned as “*probably included the free ManNAIk, metabolic intermediates, and sialylated glycans*”. Studies using bioorthogonal labeling of ManNAc derivatives with fluorescent tags introduced via click chemistry have demonstrated that labeling can occur both on the cell surface (extracellularly, *J. Am. Chem. Soc.*, 2010, 132, 16893–16899; *J. Am. Chem. Soc.*, 2013, 135, 9244–9247) and within the cell (intracellularly, *PNAS*, 2007, 104, 2614–2619). These results underscore the widespread accessibility of ManNAc derivatives across cellular compartments and emphasize the need for improved next-generation metabolic chemical reporters integrated with complementary imaging platforms.

Additionally, ManNAIk is known to label all kinds of sialylated glycoconjugates. Continuing on our previous work addressing the longstanding glycan selectivity caveat of these bioorthogonal sugars (*Front. Mol. Biosci.*, 2023, 10, 1286690.; *Carbohydr. Res.*, 2013, 377, 18–27), and the development of a new glycan selective probe (*Nat. Chem. Biol.*, 2025, 21, 681–692), we designed this target molecule in hopes of rendering it glycolipid specific. One of the reasons for this study was to design a probe that can be used to detect gangliosides using two imaging modalities. This has allowed us to reconcile the incongruities in intracellular localization seen in previous reports.

Moreover, if MM-JH-2 can be visualized using the fluorescent imaging, Raman imaging might not be necessary. What is the advantage of dual Raman/fluorescent dyes?

The reviewer notes that a previous study utilized a terminal alkyne as an SRS probe and questions the utility of a bulky phenanthrene-derivatized alkyne probe. Aside from the dual nature of the probe presented in our work, MM-JH-2 confers the advantage of being a significantly stronger Raman probe. This enables the use of spontaneous Raman spectroscopy, which allows simultaneous detection of multiple biomolecular signals, such as lipids, proteins, and nucleotides, which is not possible by SRS due to the inherently narrow spectral window in that technique. Similarly, spontaneous Raman spectroscopy enables label-free imaging of multiple classes of biomolecules, which is not possible by fluorescence microscopy.

To visualize small molecules, tiny Raman dyes are useful to avoid the decreases in bioactivity caused by introducing large fluorescent dyes. I'm afraid the introduction of large alkyne-phenanthrene tag might affect the native metabolism of mannoside, leading to the accumulation of the probe in acidic compartments.

We share the reviewer's concern that introduction of a large alkyne-phenanthrene tag might affect the native metabolism of mannosides directing them towards gangliosides, leading to the accumulation of the probe in acidic compartments. This is of course true for any modification of the metabolite and must be carefully controlled for. Our goals have been to develop more specific reagents for selectively labeling glycan species. As we show in this manuscript, the addition of the fluorescent tag both augments the Raman signature and enhances the selectivity of ganglioside labeling. We carried out many important controls to look at altered ganglioside behavior upon labeling as is outlined below:

1. Comparison with ManNAik (a known precursor of protein sialylation)
2. Pulse labeling for time-course of labeling and biosynthetic redistribution
3. Solubility characteristics of product (detergents, organic solvents)
4. In gel fluorescence of proteins showed no labeling

Why did the fluorescence not interfere with the Raman imaging? Did the authors consider the possibility that the Raman signal of alkynephenanthrene might be enhanced by resonance Raman scattering?

Fluorophores are not inherently incompatible with Raman spectroscopy, rather, the Raman excitation wavelength and detection window must only be outside the excitation

and emission windows of the fluorophore. Since MM-JH-2 is excited and emits in the blue region of the visible spectrum, a Raman excitation wavelength of 514 nm does not excite the phenanthrene ring, permitting Raman spectroscopy using this excitation wavelength. Similarly, because the laser used for Raman excitation falls outside the excitation wavelength of the phenanthrene ring there was no resonance enhancement at this wavelength. An attempt to match the absorbance wavelength of the phenanthrene ring for resonance Raman imaging would result in exactly the sort of fluorescence interference that the reviewer is concerned about.

Did the authors consider the possibility that the Raman signal of alkynephenanthrene might be enhanced by resonance Raman scattering? If the new dye proves useful for resonance Raman imaging, this work could be significant for publication. Please check the RIE, the standard of Raman intensity, in different excitation wavelengths.

MM-JH-2 was not investigated as a resonance Raman probe; the fluorescent properties of the molecule would interfere with detection of scattered photons if a resonant wavelength were used. Rather, the signal enhancement of MM-JH-2 is a result of increased π -conjugation and polarizability, as described by previous studies in model alkynes (*J. Am. Chem. Soc.*, **2012**, *134*, 20681–20689; *RSC Chem. Biol.*, **2021**, *2*, 1415-1429). MM-JH-2 confers the advantage of being a significantly stronger Raman probe. See response above for more details on the advantages of MM-JH-2 as a probe.

Reviewer #2:

This paper describes a fluorescence- and Raman-sensitive probe that specifically labels intracellular gangliosides. Gangliosides are glycolipids abundant in the nervous system and play an important role in neuronal maintenance and regeneration. Abnormal ganglioside metabolism contributes to pathological conditions such as neurodegenerative diseases and cancer, and specific visualization of gangliosides is very useful for elucidating the mechanisms of the above biological phenomena. In this study, the authors developed a mannosamine derivative (MM-JH-2) with fluorescence and Raman activity, which permeates the cell membrane and acts as a precursor for sialic acid biosynthesis in the cytoplasm and is preferentially incorporated into ganglioside biosynthesis, thereby specifically labeling gangliosides. MM-JH-2 is a novel, highly selective, and highly efficient ganglioside labeling molecule. Furthermore, MM-JH-2 has been shown to preferentially label malignant breast cancer cell gangliosides in breast cancer cells, to be highly efficient in labeling B cells and T cells, and to specifically label cancer cells at low concentrations. These results, which have been

carefully and multidimensionally validated, strongly demonstrate the utility of MM-JH-2 in biological studies: gangliosides labeled with MM-JH-2 are not only fluorescent but also Raman-sensitive, and the specific vibrational bands of alkynes MM-JH-2 can be distinguished from other intracellular biomolecules by the specific vibrational band of alkyne, and can be applied as an unprecedented and useful probe. The results presented in this paper are worthy of publication in Communication Chemistry.

We sincerely thank the reviewer for their positive review of our work.

However, to improve the quality of the paper, the following points should be considered and modified.

- 1. a discussion of why labeling with MM-JH-2 proceeds in a ganglioside-specific manner would be helpful.**

Our MALDI mass spectrometry data confirm that MM-JH-2 labels GM3 gangliosides, which contain terminal sialic acid residues. Our additional experiments also support this conclusion. MM-JH-2 fluorescence disappeared upon treatment with the lipid-dissolving detergent Triton X-100 but survived the β -hydroxy sterol-acting reagent saponin, indicating that MM-JH-2 does not label sialic acid containing glycoproteins, but rather glycolipids i.e. gangliosides. Blue fluorescence of the whole and TLC-analyzed lipid extracts from HeLa cells treated with MM-JH-2 further confirmed that the probe acts as a lipid labeling probe. In gel fluorescence scanning analysis of HeLa cell lysates treated with MM-JH-2, there was no detectable protein labeling under UV illumination light. Raman spectral imaging was used to demonstrate localization of the probe to the lipid-rich cytoplasm. Without any definite crystal structure available for GM3 synthase (ST3GAL5), it is difficult to model the enzyme with our glycan donor and describe the exact reason for this ganglioside specificity.

We have added this in the discussion section. Now the manuscript reads “Functional group modifications on the N-acetyl group of the ManNAc molecule are well tolerated by the sialic acid biosynthetic pathway, resulting in the formation of their corresponding CMP-sialic acid derivatives (*Angew. Chem. Int. Ed. Engl.*, **2016**, *55*, 9482-512). Sialyltransferase (ST) selectivity is influenced by multiple structural and biochemical factors. Key determinants include conserved protein motifs, the architecture of the catalytic site, and the length and flexibility of the ST stem region, which impacts substrate accessibility. The chemical and spatial properties of acceptor substrates further modulate enzyme specificity. Ligand-induced conformational shifts can alter ST activity and substrate affinity.

Additionally, STs form homo- and heterooligomeric complexes that facilitate substrate channeling and enhance donor and acceptor recognition. Their precise localization within cellular organelles and incorporation into supramolecular enzyme assemblies regulate substrate accessibility and catalytic efficiency. Sequence alignments of the catalytic domain identify highly conserved sialylmotifs—L (long), S (short), III, and VS (very short). Residues within motifs L and S mediate donor and acceptor substrate binding, whereas motifs III and VS contain catalytic residues critical for enzymatic activity [Rao, F. V. *et al.* Structural insight into mammalian sialyltransferases. *Nat Struct Mol Biol.*, **2009**, *16*, 1186–1188.; Kuhn, B. *et al.* The structure of human [alpha]-2,6-sialyltransferase reveals the binding mode of complex glycans. *Acta Cryst. D.*, **2013**, *69*, 1826–1838].

Our interpretation of our current results is that the hydrophobic nature of the bulky phenanthrene ring directs the MOE probe preferentially toward lipid targets rather than proteins. The lipid species which could act as acceptors are the ceramide anchored gangliosides. Moreover, the steric bulk of the CMP-phenanthrene-9-SiaAlk derivative appears to restrict its recognition to the sialylmotifs L and S of ST3Gal5, enzymes specifically involved in glycolipid biosynthesis. This observed selective incorporation into gangliosides serves to highlight important features of the largely structurally uncharacterized ganglioside specific STs like ST3GAL5.”

2. information on the analysis of molecular species of gangliosides labeled by MM-JH-2 (e.g., malignant breast cancer cells) could be added.

We thank the reviewer for raising this important point. Each cell type produces unique glycolipid species that present analytical challenges, and those studies are out of the scope of our current study. Studies investigating the specific ganglioside species labeled, as well as the pharmacological effects of MM-JH-2 treatment on malignant cells—including the detailed mechanisms underlying cancer-specific cell death—are currently ongoing and will be reported in a subsequent publication.

Reviewer #3:

This manuscript describes a dual-fluorescence Raman probe for specific biosynthetic labeling of gangliosides present in the membranes of nerve cells. This is an important topic, and significant results have been obtained. Although there are still many unknowns regarding the dynamics and functions of glycolipids within cells, the probe developed in this study is expected to lead to significant progress. However, there is a lack of sufficiently convincing explanations, and certain aspects are difficult to

understand, particularly for researchers in the field of cell biology who are interested in this study.

1. **Figure 1C: You need to explain the quantitative method. I do not understand why the ratio of MM-JH-2 staining (blue) to nuclear staining (red) exceeds 1.**

We are sorry for the confusion here. The red signal corresponds to a DNA intercalated dye which serves as a normalization control. The blue signal corresponds to specific incorporation of MM-JH-2. This ratio is expected to increase with increased labeling by MM-JH-2, and the absolute value of the ratio is somewhat arbitrary. Signal intensities exceeding a constant threshold were quantified for each channel using Fiji software. Under high concentrations of MM-JH-2 treatment, the blue channel exhibited greater above-threshold intensity compared to the red channel. Exact numerical values are provided in the source data file submitted with the manuscript.

2. **Figure 2D: Does this mean that MM-JH-2 binds to the terminal sialic acid of ganglioside? Please provide a more detailed explanation, including the relationship between MM-JH-2 binding and ganglioside specificity.**

Our MALDI mass spectrometry data directly confirms that MM-JH-2 labels GM3 gangliosides, which contain terminal sialic acid residues as the reviewer notes. NIH-3T3 cells produce predominantly GM3 gangliosides, and this is the species we have confirmed is modified. We do not exclude the possibility that other higher order gangliosides in other cell types may be labeled. Our additional experiments also support this conclusion. MM-JH-2 fluorescence disappeared upon treatment with the lipid-dissolving detergent Triton X-100 but survived the β -hydroxy sterol-acting reagent saponin, indicating that MM-JH-2 does not label sialic acid containing glycoproteins, but rather glycolipids i.e. gangliosides. Blue fluorescence of the whole and TLC-analyzed lipid extracts from HeLa cells treated with MM-JH-2 further confirmed that the probe acts as a lipid labeling probe. In gel fluorescence scanning analysis of HeLa cell lysates treated with MM-JH-2 showed no detectable protein labeling under UV illumination light. Raman spectral imaging was used to demonstrate localization of the probe to the lipid-rich cytoplasm. Without any definite crystal structure available for GM3 synthase (ST3GAL5), it is difficult to model the enzyme with our glycan donor and describe the exact reason for this ganglioside specificity.

3. Figure 3: Gangliosides are abundant in cell membranes. MM-JH-2 did not colocalize with WGA-Texas red, a cell surface marker. Why?

WGA–Texas Red binds both to sialic acids on the cell surface and to GlcNAc residues within the Golgi apparatus and within the nucleus, reflecting its broad affinity for glycan structures present throughout various cellular compartments. In contrast, our probe selectively targets gangliosides localized to lysosomes and late endosomes, enabling specific visualization of intracellular glycosphingolipid trafficking and catabolism. This differential localization highlights the distinct biological roles of these glycoconjugates and underscores the utility of our probe for studying the dynamics of ganglioside metabolism in the context of cellular homeostasis and disease. The lysosomal and late endosomal localization of MM-JH-2 is consistent with previous findings that the introduction of bulky fluorophores into the lipid moiety—or biotin substitution at the N-acetyl position of sialic acid—impairs ganglioside partitioning into ordered lipid domains of the plasma membrane. Instead, these modifications promote trafficking to acidic intracellular compartments, such as lysosomes and late endosomes (*FEBS Lett.*, 2018, 592, 3992-4006). We performed a comparative labeling experiment using the more promiscuous ManNAIk and MM-JH-2 probes, alongside colocalization analysis with WGA (Supplementary Figure 15A). The comparison of colocalization of ManNAIk, MM-JH-2, and WGA revealed cell surface colocalization between ManNAIk and WGA but not for MM-JH-2. This is consistent with the previous report suggesting a more rapid endocytosis of MM-JH-2 labeled gangliosides.

4. Is MM-JH-2 transiently labeled during the synthesis of gangliosides? Does the stability of MM-JH-2 binding to gangliosides differ depending on pH?

Under our experimental conditions (continuous feeding), MM-JH-2 labels gangliosides and reaches a steady state. This encompasses gangliosides during their biosynthesis in the Golgi apparatus, followed by rapid turnover from the plasma membrane and subsequent catabolism within acidic compartments, including lysosomes and late endosomes. A pulse-chase labeling study was performed to track MM-JH-2 localization and trafficking using the intrinsic fluorescence of the probe. Detectable fluorescence was observed within 15 minutes of MM-JH-2 treatment, with distribution consistent with labeling in the cytosol and Golgi apparatus. At 30 minutes, almost all the signal was associated with vesicle-like structures near the plasma membrane. After 1 to 2 hours fluorescence was largely localized to vesicular structures, consistent with endosomal internalization. Finally, lysosomal localization and catabolism of MM-

JH-2 was detected with prolonged incubation (16 hours). Further recycling of the probe may have commenced by 24 hours, contributing to the assembly of newly labeled gangliosides (Fig. 3C).

As the reviewer has pointed out, the stability and intensity of the MM-JH-2 signal appear to be largely dependent on intracellular pH. MALDI mass spectrometric detection of the lysosomal ganglioside further establish that the labeling is stable at intracellular pH. Notably, inhibition of intracellular acidification using Bafilomycin A1 leads to an increase in MM-JH-2 signal intensity. The lysosomal hydrolases responsible for ganglioside degradation and turnover are strongly dependent upon a low pH for their activity. For a detailed discussion, please refer to the results section and Supplementary Information Figure 13B.

5. **Considering that the labeling status of gangliosides varies depending on MM-JH-2 concentration, cell type, intracellular pH, etc., how about using a paper title that reflects this?**

We thank the reviewer for raising this excellent point. Gangliosides are itself cell type specific, so we have modified the paper title to **“A Dual Fluorescent-Raman Bioorthogonal Probe for Specific Biosynthetic Labeling of Intracellular Gangliosides”**

6. **Figure 6: It would be even better if there were images of the cells.**

We certainly agree with the reviewers idea, but unfortunately our LSRFortessa flow cytometer is not equipped to capture single cell images. We have updated Figure 6 with gated images of B and T cell populations at different concentrations of MM-JH-2 treatment.

7. **There are several instances of odd wording and grammatical errors.**

Thank you for pointing this out. We have revised the writing accordingly.

Response to the reviewer's comments:

Reviewer #1:

In the revised manuscript, I understood the unique properties of MM-JH-2, especially for the specificity for intracellular gangliosides. The Raman detection is an additional feature and not the main function of this probe. Considering the modified parts of this manuscript, this manuscript is now suitable for publication.

We sincerely thank the reviewer for their positive review of our work.

Reviewer #2:

The authors has succeeded in improving the quality of the manuscript by addressing the comments made by the reviewer. The reviewer recommend the publication of the manuscript in Commun Chem.

We sincerely thank the reviewer for their positive review of our work.

Reviewer #3:

The manuscript has been much improved.

We sincerely thank the reviewer for their positive review of our work.